# CRISPR-edited DPSCs constitutively expressing BDNF enhance dentin regeneration in injured teeth

**Ji Hyun Kim[†], Muhammad Irfan[†], Sreelekshmi Sreekumar, Atsawasuwan Phimon, Stephanie Kim, Seung Chung***

Department of Oral Biology, College of Dentistry, University of Illinois Chicago, Chicago, United States

## eLife Assessment

This study on the effect of the trophic factor BDNF upon dental cells is an understudied subject that is relevant to dental regeneration and repair. Given that the topic is new and has not been covered previously, the report is a **useful** foray into a new area of investigation, although several experimental results could be strengthened. The connection of BDNF and dental health is a **solid** attempt in potentially translating trophic factor signaling clinically, which has been stymied in past efforts.

**\*For correspondence:**
chungsh@uic.edu

[†]These authors contributed equally to this work

**Competing interest:** The authors declare that no competing interests exist.

**Abstract** Dental caries, a prevalent global health issue, results from complex bacterial interactions. In response to harmful stimuli, a desirable outcome for the tooth is the formation of tertiary dentin, a protective reparative process that generates new hard tissue. This reparative dentinogenesis is associated with significant inflammation, which triggers the recruitment and differentiation of dental pulp stem cells (DPSCs). Previously, we have demonstrated that brain-derived neurotrophic factor (BDNF) and its receptor tropomyosin receptor kinase B (TrkB), key mediators of neural functions, are activated during the DPSC-mediated dentin regeneration process. In this study, we further define the role of inflammation in this process and apply stem cell engineering to enhance dentin regeneration in injured teeth. Our data show that TrkB expression and activation in DPSCs rapidly increase during odontogenic differentiation, further amplified by inflammatory inducers and mediators such as tumor necrosis factor alpha (TNFα), lymphotoxin-alpha, and lipopolysaccharide. An in vivo dentin formation assessment was conducted using a mouse pulp-capping/caries model, where Clustered Regularly Interspaced Short Palindromic Repeats-engineered DPSCs overexpressing BDNF were transplanted into inflamed pulp tissue. This transplantation significantly enhanced dentin regeneration in injured teeth. To further explore potential downstream pathways, we conducted transcriptomic profiling of TNFα-treated DPSCs, both with and without TrkB antagonist cyclotraxin-B. The results revealed significant changes in gene expression related to immune response, cytokine signaling, and extracellular matrix interactions. Taken together, our study advances our understanding of the role of BDNF in dental tissue engineering using DPSCs and identifies potential therapeutic avenues for improving dental tissue repair and regeneration strategies.

## Introduction

Dental caries, commonly known as tooth decay, remains one of the most prevalent chronic diseases, affecting a significant portion of the global population (*Peres et al., 2019*). This condition results from the complex interplay between bacterial biofilms, dietary sugars, and host factors, leading to the demineralization of tooth enamel and dentin (*Bowen et al., 2018*; *Mosaddad et al., 2019*). In

recent years, there has been growing interest in the role of dental pulp stem cells (DPSCs) in regenerative dentistry due to their ability to differentiate into dentin-forming odontoblast-like cells and their potential in tooth repair and regeneration (*Potdar and Jethmalani, 2015*). It is widely recognized that the regeneration of the dental-pulp complex is closely associated with inflammation, especially since dental caries and the subsequent regenerative processes take place within an inflammatory environment (*Galler et al., 2021*). Inflammation serves as a vital biological response for ensuring host survival in the face of infection and tissue injury, playing a crucial role in maintaining normal tissue homeostasis (*Peiseler and Kubes, 2018*). It is generally accepted that the regeneration of the injured tooth is closely tied to inflammation (*Lin and Rosenberg, 2011*). Specifically, it plays a pivotal role in the regeneration of injured dental tissues by stimulating the recruitment, proliferation, and differentiation of pulp progenitor cells and odontoblasts (*Nakashima et al., 2013*). However, an imbalance between inflammation and repair can lead to irreversible tissue damage. While inflammation is recognized as necessary for proficient tissue regeneration, only a few studies have explored its role in dentin regeneration.

The brain-derived neurotrophic factor (BDNF) and its receptor, tropomyosin receptor kinase B (TrkB), constitute a critical signaling pathway involved in the survival, development, and function of neurons (*Ferrer et al., 1999*; *Huang and Reichardt, 2003*; *Zhang et al., 2012*). Emerging evidence suggests that the BDNF-TrkB pathway also plays a significant role in the inflammatory response and tissue repair processes (*Cappoli et al., 2020*; *Hang et al., 2021*). The overexpression of BDNF in DPSCs has been shown to enhance their regenerative capabilities, which may be particularly beneficial in the context of dental caries where inflammation and tissue damage are prominent. Indeed, our previous study demonstrated the critical role of BDNF/TrkB in the DPSC odontoblastic differentiation under inflammation (*Kim et al., 2025*). Tumor necrosis factor alpha (TNFα)-stimulated BDNF/TrkB showed an enhanced odontoblast differentiation of DPSCs.

Although the general principles for successful BDNF-induced regeneration have been proposed, achieving clinical success remains challenging. One significant obstacle is the clinical application of BDNF itself as the recombinant protein has an extremely short half-life, which greatly diminishes its effectiveness (*Miranda-Lourenço et al., 2020*). Therefore, establishing a stable and continuous BDNF production platform is essential, and stem cell engineering could potentially address this critical need in the future. Considering the essential role of BDNF/TrkB signaling activation in the inflammatory dentinogenesis observed in cell culture conditions, there is a strong case for validating BDNF/TrkB data in vivo. In this study, we further explore the role of various inflammatory agents in modulating BDNF receptor, TrkB, as well as the potential therapeutic applications of BDNF overexpression in DPSCs using Clustered Regularly Interspaced Short Palindromic Repeats (CRISPR) technology for treating dental caries. By uncovering the mechanisms through which the BDNF-TrkB pathway affects DPSC function and inflammation, we aim to offer insights into innovative regenerative strategies that could enhance dental health outcomes.

## Materials and methods
### Chemicals and reagents

Human DPSCs were purchased from Lonza, Pharma & Biotech (Cat# PT-5025). MEM-alpha, DMEM, phosphate buffer solution (PBS), fetal bovine serum, L-glutamine, and Antibiotic–Antimycotic were procured from Gibco Fisher Scientific (Waltham, MA, USA). Poly-D-lysine-coated (BioCoat, 12 mm) round German glass coverslips were purchased from Corning Fisher Scientific (Cat# 354087; Waltham). RIPA buffer was from Cell Signaling Technology (Danvers, MA, USA). Various antibodies were procured: anti-TrkA (RRID:AB_354970, R&D Systems Cat# AF175), anti-phospho-TrkA (R&D Systems Cat# AF5479), anti-TrkB (RRID:AB_2686965, BioLegend Cat# 695102), anti-phospho-TrkB (RRID:AB_2553666, Invitrogen Cat# PA5-36695), pro-BDNF (5H8) (RRID:AB_1128219, Cat# sc-65514), β-tubulin (RRID:AB_2537819, Invitrogen Cat# MA5-16308), anti-β-actin (RRID:AB_2539914, Invitrogen Cat# PA1-183) from Fisher Scientific (Waltham), anti-GFP (Santa Cruz Biotechnology Cat# sc-8334, RRID:AB641123), and anti-STRO-1 (Santa Cruz Biotechnology Cat# sc-7612, RRID:AB2113895) from Santa Cruz (Dallas, TX, USA). Chameleon Vue pre-stained protein ladder (Cat. #. P/N: 928-50000) was from LI-COR bio. Fluorescent secondary antibodies were from Life Technologies (Grand Island, NY, USA). TNFα was from Invitrogen, Fisher Scientific (Waltham), and a few other chemicals were from

Fisher Chemical (Nazareth, PA, USA). BDNF CRISPR activation plasmid (h) (Cat# sc-400029-ACT) and Reagent System were purchased from Santa Cruz Biotechnology (Dallas).

## Cell culture

All experiments were conducted with different sets of human DPSCs 3–4 times, using cells at the second and third passages, and cell proliferation was measured by counting the total number of cells (occasionally tested for mycoplasma contamination using MycoStrip, InvivoGen). Commercially available human DPSCs isolated from third molars of adult donors (ages 29–30 years old) collected during the extraction of wisdom teeth, which were guaranteed and authenticated through 10 population doublings, to express CD105, CD166, CD29, CD90, and CD73, and do not express CD34, CD45, and CD133 106-108; cultured in regular growth media (αMEM containing 10% FBS, 1% L-glutamine, and 1% antimycotic/antibiotic) at 37°C and 5% $CO_2$ for 3–4 days. After this initial period, the media was swapped to odontogenic media (DMEM containing 10% FBS, 1% L-glutamine and antimycotic/antibiotic, 50 µg/mL ascorbic acid, 10 mM β-glycerophosphate, and 10 nM dexamethasone) at day 4 until day 17. TrkB antagonist (cyclotraxin-B [CTX-B], 200 nM/mL) was treated every 3 days until day 17. Cells were treated with lipopolysaccharide (LPS), TNFα, lymphotoxin-alpha (LTA), C5a, and IL-5 at days 4 and 7. The CaMKII inhibitor (5 µM/mL) or CaMKII protein (1 µM/mL) was treated with dentinogenic media at days 4, 7, 10, and 14.

## Quantitative real-time PCR analysis of odontogenic differentiation marker gene expression in dental pulp stem cells

DPSCs were cultured in a 6-well plate at $5×10^4$ cells per well. The total mRNA was extracted using an RNeasy Mini Kit (74104, QIAGEN), and the containing quantity of cDNA was measured using the NanoDrop 2000 (ND2000, Fisher Scientific). The Fast SYBR Green Master Mix (4385616, ThermoFisher) was used to identify the cDNA sample according to the manufacturer's protocol. Primer sequences (Integrated DNA Technologies) were used predesigned *GAPDH*. (forward: 5'-GGC ATC CAC TGT GGT CAT GAG-3'; reverse: 5'-TGC ACC ACC AAC TGC TTA GC-3'), *DSPP* (forward: 5'-CTG TTG GGA AGA GCC AAG ATA AG-3'; reverse: 5'-CCA AGA TCA TTC CAT GTT GTC CT-3'), and *DMP-1* (forward: 5'-CAC TCA AGA TTC AGG TGG CAG-3'; reverse: 5'-TCT GAG ATG CGA GAC TTC CTA AA-3').

## Immunocytochemistry

The seeded DPSCs were incubated in a 12-well plate with coverslips at 37°C in a $CO_2$ incubator until reaching 70–80% cell confluency. The coverslips were then fixed with 4% paraformaldehyde for 2 h at 4°C. Blocking and permeabilization were performed using 5% normal goat serum (NGS) and 0.01% Triton X in 0.01 M PBS for 1 h at room temperature (RT). For primary antibody treatment, the specimens were treated overnight at 4°C with the following antibodies diluted in 5% serum: anti-TrkB (1:1000), anti-pTrkB (1:1000), and/or anti-STRO1 (1:1000). After the overnight incubation with primary antibodies, the secondary antibodies were applied for 2 h with Alexa Fluor-594 anti-mouse IgG, Alexa Fluor-488 anti-rabbit IgG (1 µg/mL), and/or DAPI (1 µg/mL). The coverslips were mounted on glass slides, and images were taken using a Leica microscope. MATLAB (R2022a) software was used to measure fluorescence intensity and perform quantification.

## BDNF CRISPR activation plasmid transfection in hDPSCs

DPSCs were incubated in a 6-well plate with 3 mL of antibiotic-free standard growth medium at 37°C in a 5% $CO_2$ incubator until reaching 40–60% confluency in preparation for CRISPR activation plasmid transient transfection. For the preparation of transfection solutions, 2 µg of plasmid DNA was diluted in Plasmid Transfection Medium (sc-108062) to a final volume of 150 µL (Solution A) and mixed and incubated for 5 min at RT. In parallel, 10 µL of UltraCruz Transfection Reagent (sc-395739) was diluted in Plasmid Transfection Medium (sc-108062) to a final volume of 150 µL (Solution B) and incubated for 5 min at RT. Solutions A and B were then combined by vortexing immediately and incubated for 20 min at RT. For the transfection procedure, the cell culture medium was replaced with a fresh antibiotic-free growth medium, and 300 µL of the Plasmid DNA/UltraCruz Transfection Reagent Complex was added to each well. Details regarding BDNF CRISPR Activation Plasmid (h) protocol are as follows: BDNF CRISPR Activation Plasmid (h) is a synergistic activation mediator (SAM) transcription activation system

designed to upregulate gene expression specifically. BDNF CRISPR Activation Plasmid (h) consists of three plasmids at a 1:1:1 mass ratio: a plasmid encoding the deactivated Cas9 (dCas9) nuclease (D10A and N863A) fused to the transactivation domain VP64, and a blasticidin resistance gene; a plasmid encoding the MS2-p65-HSF1 fusion protein, and a hygromycin resistance gene; a plasmid encoding a target-specific 20 nt guide RNA fused to two MS2 RNA aptamers, and a puromycin resistance gene. The resulting SAM complex binds to a site-specific region approximately 200–250 nt upstream of the transcriptional start site and provides robust recruitment of transcription factors for highly efficient gene activation. The cells were incubated for 48 h under standard culture conditions. Protein analysis was conducted to confirm overexpression. Following transfection, WB assayed gene activation efficiency by using pro-BDNF antibody (5H8): sc-65514.

## Western blot

To verify BDNF activation using CRISPR plasmid, cell lysates were prepared using RIPA buffer (50 mM Tris pH 7.6, 150 mM NaCl, 1% Triton X100, 1 mM EDTA, 0.5% sodium deoxycholate, 0.1% SDS) containing protease inhibitor (Roche, Indianapolis, IN, USA) and protein was measured using BCA method (Pierce BCA Protein Assay Kit, Thermo Fisher Scientific, Lenexa, KS, USA). Equal amounts of protein (35 µg) were loaded and separated by SDS-polyacrylamide gel electrophoresis and transferred onto nitrocellulose (Bio-Rad, CA, USA). Blots were probed using polyclonal antibodies specific for pro-BDNF or β-actin overnight at 4°C and then washed and probed with secondary antibodies for 2 h at RT. Odyssey CLx visualized the protein bands.

## In cell western assay

Human DPSCs were seeded in growth media at $15 \times 10^3$ cells/cm$^2$ in 96-well optical plates. At sub-confluency, cells were incubated in an antibiotic-free medium and treated as mentioned above. Then, cells were immediately fixed with 100% cold methanol (15 min) and saturated with 5% BSA (1.5 h). Cells were incubated overnight at 4°C, anti-TrkA, anti-TrkB, anti-phospho-TrkA, anti-phospho-TrkB, or anti-β-tubulin. Cells were then washed (0.05% Tween-20/PBS) and incubated with respective IRDye-680RD or IRDye-800RD secondary antibody (1 h) at RT. After five washes, plates were dried and scanned at 700 and/or 800 nm (Odyssey CLx).

## Animals

The Institutional Animal Care and Use Committee at the University of Illinois Chicago reviewed and approved all surgical and experimental procedures (protocol no.: 21-197). C57BL/6 male mice, 6–8 weeks old (n=12), were purchased from Jackson Laboratory (#000461) and housed three per cage in a temperature-controlled room (23±1°C, 12 h/12 h light/dark cycle) and maintained at the UIC animal facility. After 1 week of acclimatization, mice were randomly selected and divided into two groups (control and transplantation; n=6 each) for experimentation. For the molars, the mandibles of the mice were collected for further experimentation.

## Transplantation of CRISPR-engineered BDNF-overexpressing DPSC in the pulp-capping mouse model

C57BL/6 male mice aged 6–8 weeks were employed for this mouse model. Dentin of the lower left first molar was penetrated using a 0.3 mm rounded carbide burr drill operating at automatic speed. Subsequently, an 11G needle was utilized to remove any remaining dentin upon exposure. The exposed dentin was then treated with BDNF CRISPR Activation Plasmid-transfected DPSCs embedded in collagen (from rat tail), followed by sealing with mineral trioxide aggregate to shield the pulp from further inflammation. Animals were euthanized beginning 4 weeks post-injury, and their mandibles were excised. The molars were subsequently harvested for histological, molecular, and immunohistochemical analyses to evaluate the efficacy of the treatment in promoting dental pulp regeneration from the person blinded to the study.

## Micro-CT (µCT)

Mandibles from C57BL/6 male mice were collected from both the sham control group and the group receiving transplantation of CRISPR-engineered BDNF-overexpressing DPSCs in the pulp-capping mouse model. µCT analyses were conducted using the commercial service at RUSH Hospital to assess

the volume of regenerated dentin quantitatively. All animal phenotype analyses were blinded to mitigate potential examiner bias and ensure impartial evaluation.

## H&E staining

The mandibles of C57BL/6 mice were harvested and fixed in a 4% formaldehyde solution immediately after extraction. Subsequently, they were placed in decalcification solution within 15 mL Falcon tubes for 2 days, with daily changes in concentration (starting from 10% sucrose in PBS, progressing to 20% and then 30%). The mandibles were sectioned following decalcification, and histological staining protocols were optimized specifically for dental tissues. Sections underwent a sequence of staining procedures, including bleaching, eosin staining, and dehydration through graded ethanol solutions (ranging from 70% to 100% ethanol). Finally, the sections were cleared in xylene to prepare them for microscopic analysis and further histopathological evaluation.

## Collection of human virgin teeth and caries teeth

Human wisdom tooth extractions were conducted by licensed dentists specializing in oral surgery. This study involving human specimens received ethical approval from the Endodontics clinic at the College of Dentistry, University of Illinois Chicago (protocol no.: 2012-0588). Informed written consent was not required due to the nature and objectives of the experimental procedures. Ethical considerations and patient confidentiality were strictly followed throughout the study, ensuring compliance with established guidelines and protocols for human research.

## Immunohistochemistry

The extracted mandibles were embedded in an OCT cryostat sectioning medium and stored at –20°C for 2 h before being cut into 20 µm thick sections. The mounted tissues were fixed with 4% paraformaldehyde for 1 h at RT. Subsequently, blocking and permeabilization were performed using 10% NGS and 0.1% Triton X in 0.01 M PBS for 1 h at RT. For primary antibody incubation, the specimens were treated overnight at 4°C with mouse anti-CaMKII (1:200), rabbit anti-p-CaMKII (1:200), mouse anti-GFAP (1:200), and/or mouse anti-STRO1 (1:200), diluted in 10% NGS. Following overnight incubation with primary antibodies, the specimens were incubated for 2 h at RT with secondary antibodies: Alexa Fluor-594 anti-mouse IgG, Alexa Fluor-488 anti-rabbit IgG (1 µg/mL), and/or DAPI (3 µg/mL). After secondary antibody treatment, the specimens were mounted on glass slides, and images were acquired using a Leica microscope. Fluorescence intensity was quantified using ImageJ software. Representative images were exported in TIFF format. ImageJ software and GraphPad Prism version 10 were employed for fluorescence intensity analysis.

## RNA sequencing (poly-A RNA seq)

RNA sequencing was employed to comprehensively analyze the transcriptome, focusing specifically on mRNA, utilizing services provided by LC Biosciences (Houston, TX, USA). This method capitalizes on the polyadenylated (poly-A) tails present at the 3' ends of eukaryotic mRNA molecules. DPSCs underwent a 10-day differentiation protocol with or without specific treatments. Post-differentiation, total RNA was extracted from the odontogenic differentiated DPSCs for sample preparation. LC Biosciences conducted rigorous assessments to evaluate the extracted RNA's quantity and quality, followed by poly-A selection to enrich for mRNA transcripts. Subsequent cDNA synthesis facilitated library preparation, paving the way for high-throughput sequencing to explore the transcriptomic landscape in depth. This approach enabled the identification and characterization of gene expression patterns associated with the differentiation of DPSCs under various experimental conditions.

## Statistical analysis

The statistical analyses were performed on at least three independent experiments with duplicates or triplicates, and statistical significance was determined using one-way analysis of variance followed by post hoc Dunnett's test (SAS 9.4) to compare the different treatments and their respective controls (p-value of 0.05 or less was considered statistically significant). In addition, the data were analyzed using Tukey's test to determine statistical significance between the groups. For quantification of immunofluorescence staining intensity, ImageJ 1.49v software was used. Fixed areas of 1 mm × 1 mm or 2 mm × 2 mm were selected to analyze differentiated cells' number or fluorescence intensity.

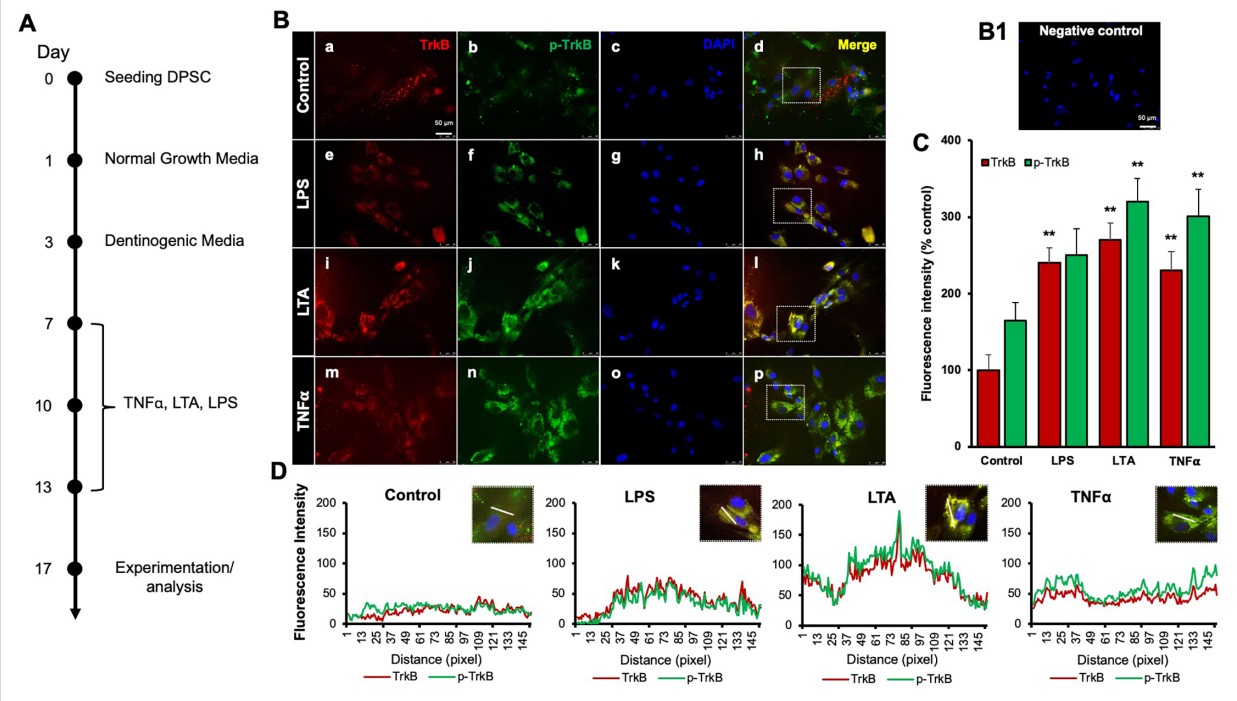

**Figure 1.** Effects of various inflammatory mediators on the expression of tropomyosin receptor kinase B (TrkB) and its phosphorylation in human dental pulp stem cells (DPSCs). (**A**) Schematic representation of DPSC differentiation stimulated by various inflammatory mediators. (**B**) DPSCs were cultured in normal growth media with or without lipopolysaccharide (LPS), lymphotoxin-alpha (LTA), or tumor necrosis factor alpha (TNFα) for 24 h, and cells were fixed, and double immunofluorescent staining was performed (TrkB: red; p-TrkB: green). Cells were counter-stained with DAPI (blue). (**a–d**) The control cells showed less TrkB and p-TrkB expression than the LPS (**e–h**), LTA (**i–l**), or TNFα (**m–p**) treated groups. (**B1**) Negative control. (**C**) The bar graph shows a significant increment in TrkB and p-TrkB expression in the groups treated with inflammatory mediators. *p<0.05 and **p<0.01 vs. control. (**D**) The line graph shows that the co-localization of TrkB and p-TrkB are higher peaks observed in LPS, LTA, and TNFα-treated groups than in control. Scale bar: 50 μm.

## Results

### Various inflammatory mediators increase TrkB expression and activation in DPSCs

Our previous study demonstrated that TNFα augments the odontoblast differentiation of DPSCs in a TrkB-dependent manner (*Irfan and Chung, 2023*; *Kim et al., 2023b*). The literature indicates that pro-inflammatory cytokines, such as TNFα, LTA, and LPS, play significant roles in the formation of tertiary dentin (*Irfan et al., 2022b*). To further confirm the role of inflammation in this process, various inflammatory mediators were applied during DPSC odontoblastic differentiation. The outline of the experimental timeline is shown (*Figure 1A*). Human DPSCs were seeded on day 0, followed by culture in normal growth media. On day 3, the cells were transferred to dentinogenic media. TNFα, LTA, and LPS treatments were initiated every other 3 days. The representative image shows the immunofluorescence staining results for TrkB and phosphorylated TrkB (p-TrkB) under various treatments (*Figure 1B*). The quantification of fluorescence intensity across these different treatment groups was analyzed by percentages of control (*Figure 1C*). TrkB expression and activation in DPSCs were significantly upregulated during odontogenic differentiation, especially under inflammatory stimulants such as TNFα, LTA, and LPS by 301±17, 320±15.2, and 250±19, respectively, vs. control 165±12.4 (p<0.01). The spatial distribution of fluorescence intensity for TrkB and p-TrkB was measured across different treatment groups (*Figure 1D*). Figures were analyzed using ZEN (blue) software, and graphs were created. This spatial analysis confirms the enhanced and localized expression of TrkB and p-TrkB in response to inflammatory stimuli.

Next, we examined the expression levels of TrkA and TrkB receptors in response to various inflammatory stimuli in normal and dentinogenic media. TrkA and TrkB activate overlapping but distinct downstream signaling pathways, leading to different cellular outcomes. Depending on context, their

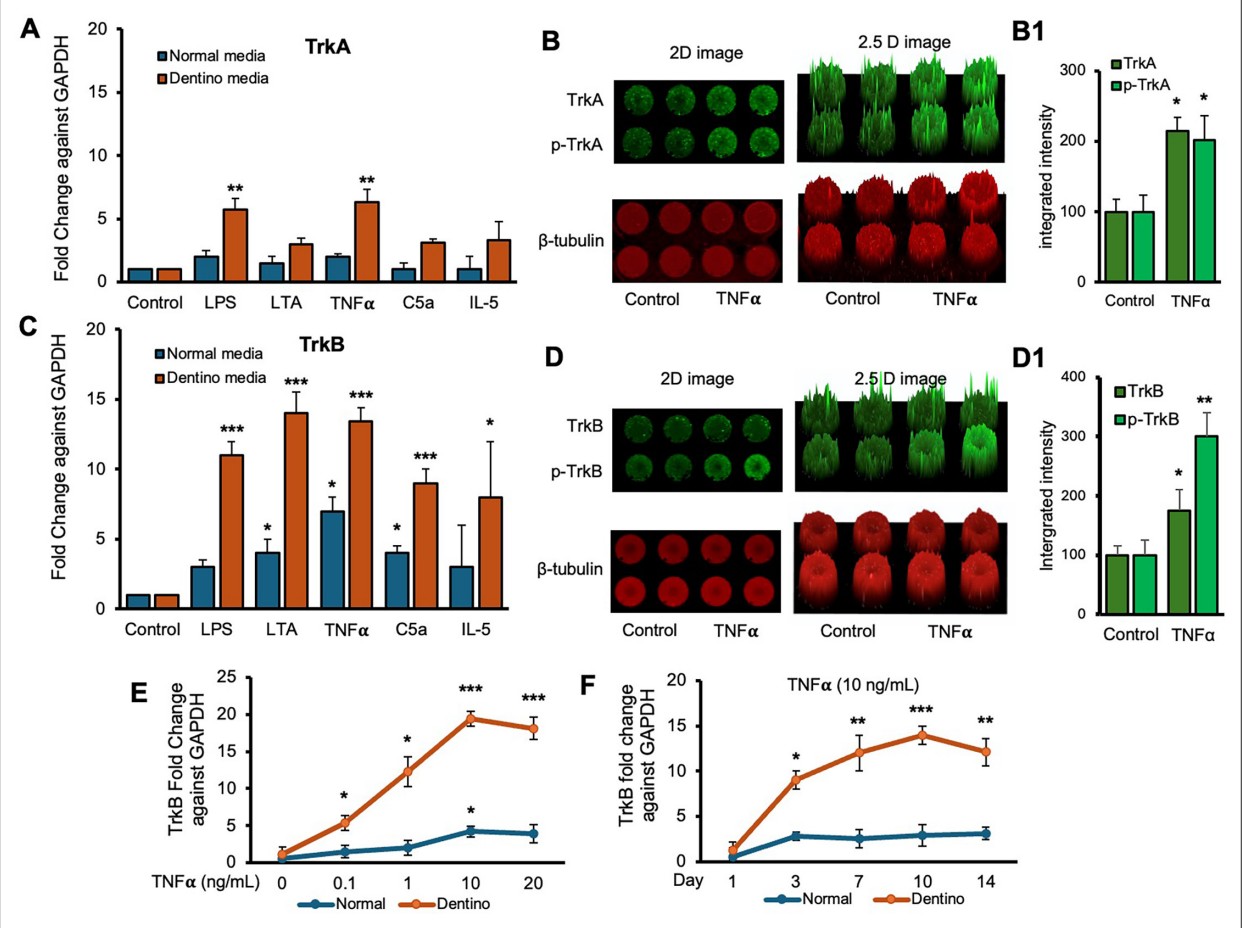

**Figure 2.** Time-course analyses of tropomyosin receptor kinase B (TrkB) mRNA expression in dental pulp stem cells (DPSCs). (**A, C**) Real-time PCR was used to determine the effect of various inflammatory mediators (lipopolysaccharide [LPS], lymphotoxin-alpha [LTA], interleukin 5 (IL5), tumor necrosis factor alpha [TNFα], and C5a) on the expression level of TrkA and TrkB mRNA in DPSCs after 1-day treatment in regular growth medium (blue bars) or dentinogenic medium (red bars). While these inflammatory components do not affect much the expression of TrkA or TrkB in normal growth medium, inflammatory mediators such as LPS, LTA, and TNFα stimulation increase TrkB expression in DPSCs undergoing odontoblastic differentiation (orange bars 'LPS, LTA, and TNFα' vs. orange bar 'Control'). Comparatively, TrkA expression changed little during the odontoblastic differentiation of DPSCs (**A**). (**B, D**) In cell western assay showing effects of TNFα on the expression of TrkA and TrkB and their phosphorylation in 2D and 2.5 D models. Graphs (**B1, D1**) show fluorescence intensity TNFα vs. control. (**E**) Effect of TNFα on TrkB expression. DPSCs were cultured in regular or dentinogenic/osteogenic medium and with 0, 0.1, 1, 10, or 20 ng/mL of TNFα. TrkB mRNA expression was determined by real-time PCR analysis after 24 h of culture. TNFα stimulation, independent of the concentration used, quickly potentiates the odontogenesis-regulated expression of TrkB. However, this TrkB expression increase is sustained with increasing concentrations of TNFα treatment, even at 20 ng/mL. (**F**) Real-time PCR was used to evaluate the level of TrkB mRNA in DPSCs after 1, 3, 7, 10, and 14 days of culture in regular medium (blue lines) or dentinogenic medium (orange lines) in TNFα-stimulated DPSCs odontoblastic differentiation. In contrast to relatively stable TrkB levels in undifferentiated DPSCs, TrkB expression quickly increased in dentinogenic medium, with the most significant increase detected between 7 and 10 days (fold increase: 7 day = 12.04 ± 2.1, 10 days = 14.1 ± 1.72). Results are expressed as relative expression to glyceraldehyde 3-phosphate dehydrogenase (*GAPDH*). Data are presented as mean ± SD of three independent experiments. *p<0.05 vs. regular medium, *p<0.05, and **p<0.01, ***p<0.001 vs. dentinogenic medium and without inflammatory stimulation.

differential expression and activation patterns can influence cell survival, apoptosis, or differentiation. Therefore, comparing TrkA and TrkB is essential to elucidate neurotrophin-specific effects in a given experimental model. The results indicate significant differential expression of these receptors under both conditions. TrkB expression was modulated by the media and stimuli (*Figure 2C*). Under control conditions, no significant differences were observed between the media types. However, LPS, LTA, TNFα, and C5a significantly increased TrkB expression in dentinogenic media compared to regular media (p<0.001 for LPS, TNFα, and C5a; p<0.05 for LTA). IL-5 caused a slight but significant increase in TrkB expression in dentinogenic o media (p<0.05). Conversely, baseline TrkA expression showed no

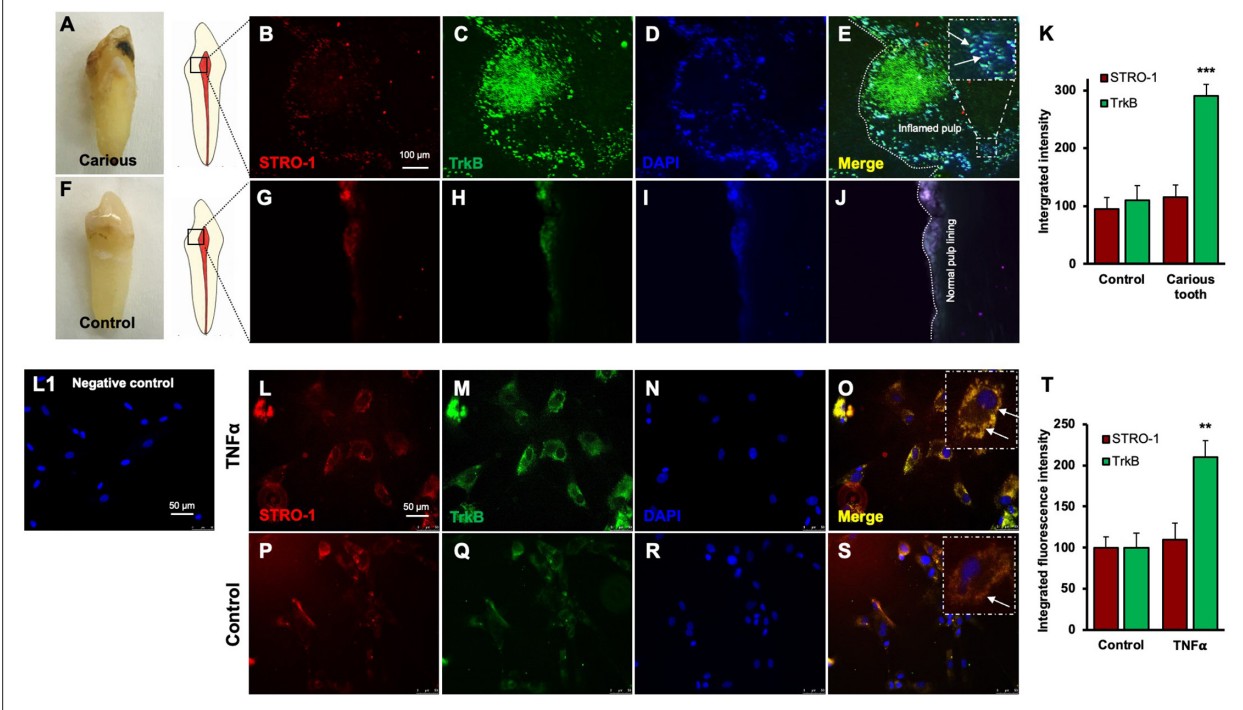

**Figure 3.** Expression of STRO-1 and tropomyosin receptor kinase B (TrkB) in human carious or normal tooth in vivo and dental pulp stem cells (DPSCs) in vitro. (**A–J**) Immunofluorescence double staining was used to localize STRO-1 and TrkB-expressing cells in the human virgin (control) and carious tooth section. While the expression of STRO-1 was detected in the vicinity of blood vessels, TrkB is expressed more in inflamed pulp tissue compared with virgin tooth. Merged images of STRO-1 and TrkB staining demonstrated a co-expression of TrkB and STRO-1 in the perivascular area. Scale bar: 100 μm. Nuclei were counterstained with DAPI (blue). (**L1**) Negative control (**K**) The bar graph shows a significantly higher expression of TrkB in the carious tooth section (***$p < 0.001$ vs. control). (**L–S**) Expression of STRO-1 and TrkB by DPSCs in vitro. Immunofluorescence double staining was used to analyze TrkB expression in control or untreated and TNFα-treated cells. All control DPSCs express both the mesenchymal stem cell marker STRO1 and TrkB, while TNFα-treated cells' TrkB expression was significantly higher than control. Merged images of STRO1 and TrkB revealed that all STRO-1-positive cells expressed TrkB. Nuclei were counterstained with DAPI (blue). Scale bar = 50 μm. (**T**) The bar graph shows significantly higher expression of TrkB in TNFα-treated cells ($p < 0.01$ vs. control).

significant difference between regular and dentinogenic media (*Figure 2A*), and C5a and IL-5 did not induce significant changes in TrkA expression in either medium.

*Figure 2B and D* show the effect of TNFα stimulation on the expression of TrkA and TrkB and their phosphorylation vs. control. 2D and 2.5 models show fluorescence spikes to show the phenomenon. TNFα stimulation caused significant phosphorylation of TrkB ($p < 0.01$) compared to TrkB ($p < 0.05$). Graphs (*Figure 2B1 and D1*) show significant changes among TNFα-treated cells vs. control. So, we chose TrkB as a target receptor for our further study.

The dose-dependent response of TrkB to TNFα (*Figure 2E*) revealed minimal baseline expression at 0 ng/mL TNFα, with significant increases at 0.1 ng/mL, 1 ng/mL, and a pronounced peak at 10 ng/mL in dentinogenic media ($p < 0.001$) compared to 0.1 and 1 ng/mL. At 20 ng/mL TNFα, TrkB expression remained significantly higher in dentinogenic media than in normal media ($p < 0.001$). The time-dependent response to a fixed TNFα concentration (10 ng/mL) showed that TrkB expression in dentinogenic media was significantly higher on day 1, further increased by day 3, and peaked on day 7 ($p < 0.001$) (*Figure 2F*). Although TrkB levels slightly decreased on days 10 and 14, they remained significantly elevated compared to normal media. These findings emphasize the importance of the local microenvironment in influencing receptor expression and inflammatory responses.

## TrkB expression is increased in human carious teeth and in DPSCs

Regenerative odontogenic differentiation in the context of caries occurs within an inflammatory environment, making it essential to understand the impact of inflammation on DPSCs to comprehend

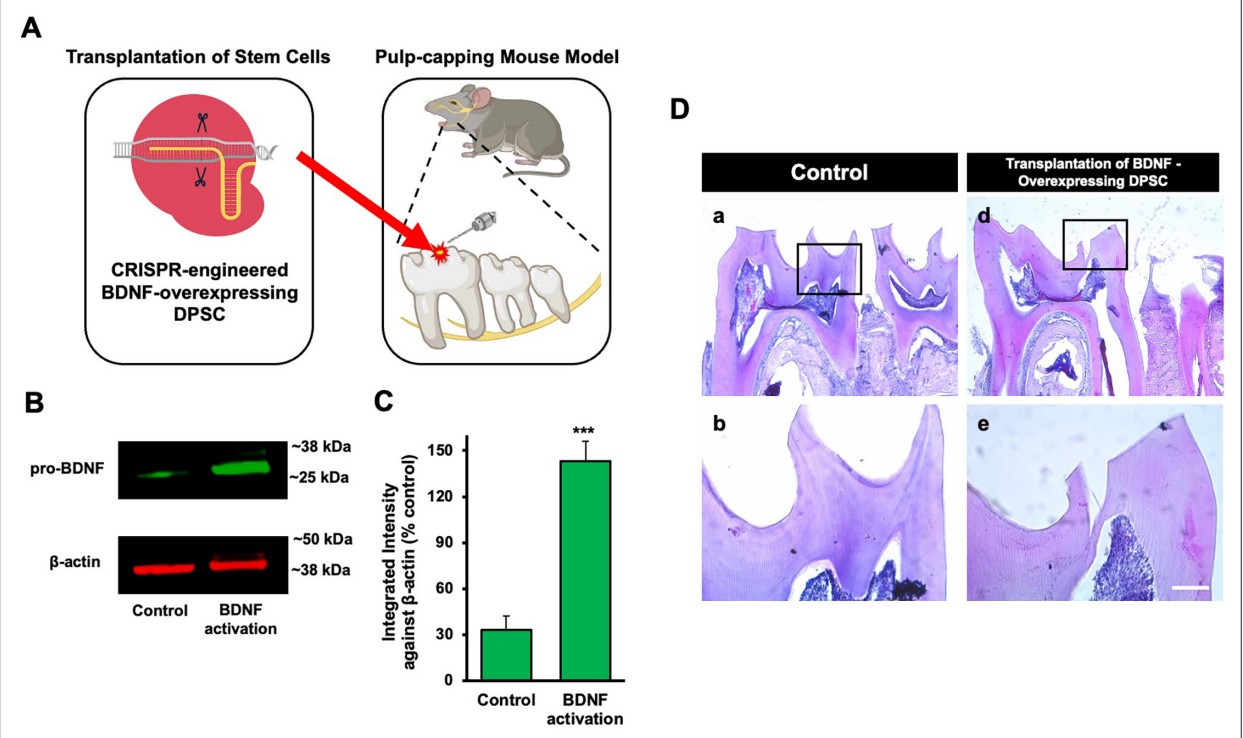

**Figure 4.** The transplantation of Clustered Regularly Interspaced Short Palindromic Repeats (CRISPR)-engineered brain-derived neurotrophic factor (BDNF)-overexpressing dental pulp stem cells (DPSCs) in the pulp-capping mouse model. (**A**) A schematic representation of the transplantation of DPSC into the first molar tooth after drilling. (**B**) The confirmation of BDNF CRISPR activation plasmid enhanced the expression of pro-BDNF. (**C**) Bar graph showing the integrated intensity of CRISPR-engineered BDNF-activated DPSCs against β-actin compared to control. (**D**) H&E staining of sham control and injured tooth in mouse (n = 6 each group) transplanted with CRISPR-engineered BDNF-overexpressing DPSCs. Scale bar: 100 μm.

The online version of this article includes the following source data for figure 4:

**Source data 1.** Original files for western blot images displayed in *Figure 4B*.

**Source data 2.** Original files for western blot images displayed in *Figure 4B* with labeling.

dentin regeneration fully (*Kim et al., 2023a*). Immunofluorescence double staining was employed to analyze the expression of STRO-1, a stem cell marker (*Lin et al., 2011*), and TrkB in human carious and normal teeth, as well as in DPSCs (*Figure 3A–K*). The in vivo analysis revealed that STRO-1 was primarily localized around blood vessels, while TrkB was more abundantly expressed in the inflamed pulp tissue of carious teeth compared to normal teeth. Quantitative analysis revealed significantly higher TrkB expression in carious teeth ($p<0.001$). In vitro, both STRO-1 and TrkB were expressed in control DPSCs, but TNFα treatment markedly increased TrkB expression (*Figure 3L–T*). The data showed that TrkB expression was significantly elevated in TNFα-treated DPSCs ($p<0.01$). These findings suggest that TrkB is upregulated in response to inflammation in both carious teeth and DPSCs.

## Transplantation of GFP-tagged CRISPR-engineered BDNF-overexpressing DPSCs enhances dentin regeneration in a pulp-capping mouse model

Our previous data demonstrated that inflammation and BDNF-induced DPSC odontogenic differentiation increased the expression of odontogenic differentiation markers (*Kim et al., 2023b*). To investigate whether BDNF-TrkB signaling controls dentin regeneration following inflammation in vivo, we utilized CRISPR-engineered BDNF-overexpressing DPSCs in a pulp-capping mouse model (*Figure 4A*). We initially utilized the CRISPR technique to activate BDNF expression in DPSCs. Transfection of DPSCs with the BDNF CRISPR activation plasmid was confirmed through western blot analysis, which revealed increased BDNF expression compared to the control group (*Figure 4B*). Quantification of the integrated intensity relative to β-actin further indicated a significant increase in BDNF expression

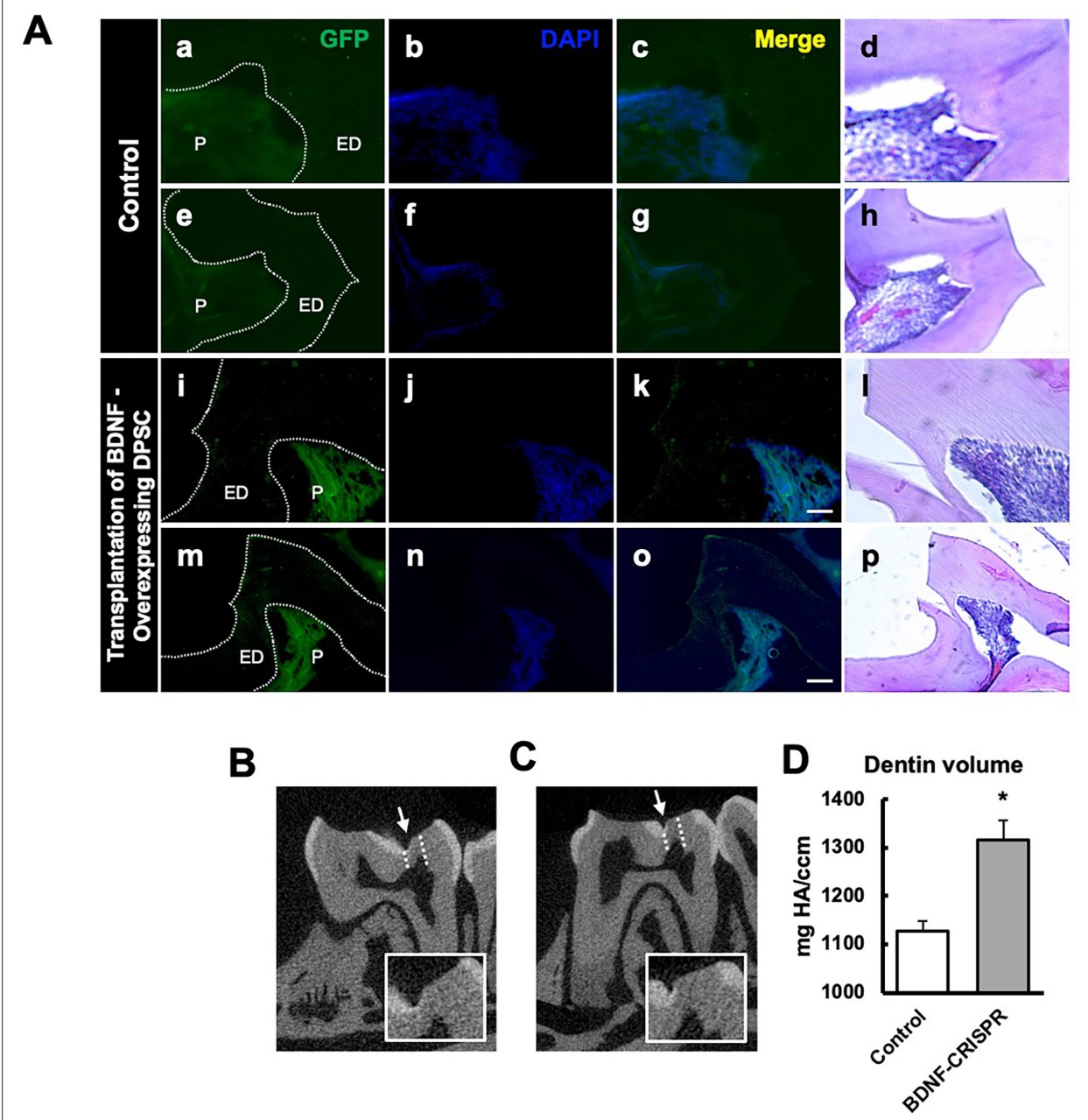

**Figure 5.** The transplantation of Clustered Regularly Interspaced Short Palindromic Repeats (CRISPR)-engineered brain-derived neurotrophic factor (BDNF)-overexpressing dental pulp stem cells (DPSCs) in the pulp-capping mouse model and micro-CT analysis. (**A**) Immunohistochemistry was performed to assess GFP-tagged transplanted cells. The white arrow indicates the pulp lining. Scale bar: 100 μm (**B**) Micro-CT image in the sham control of the pulp-capping mouse model. The white arrow indicates the drilling operation, and the dotted line specifies the injured area of dentin. The white box is a magnified image of the injured area. (**C**) The transplantation of CRISPR-engineered BDNF-overexpressing DPSCs in the pulp-capping mouse model. (**D**) Analyzed density of dentin compared with sham control vs. transplantation of CRISPR-engineered BDNF-overexpressing DPSCs in the pulp-capping mouse model (n=5). p<0.05 vs. control.

in CRISPR-activated DPSCs (*Figure 4C*). Histological analysis of H&E-stained sections was performed to evaluate tissue morphology and surgery site from the transplantation of CRISPR-engineered BDNF-overexpressing DPSCs in a pulp-capping mouse model (*Figure 4D*).

Following confirmation of successful CRISPR activation, we conducted the transplantation of CRISPR-engineered BDNF-overexpressing DPSCs on dentin regeneration in a pulp-capping mouse model. Micro-CT imaging was utilized to assess dentin volume in both sham control and experimental

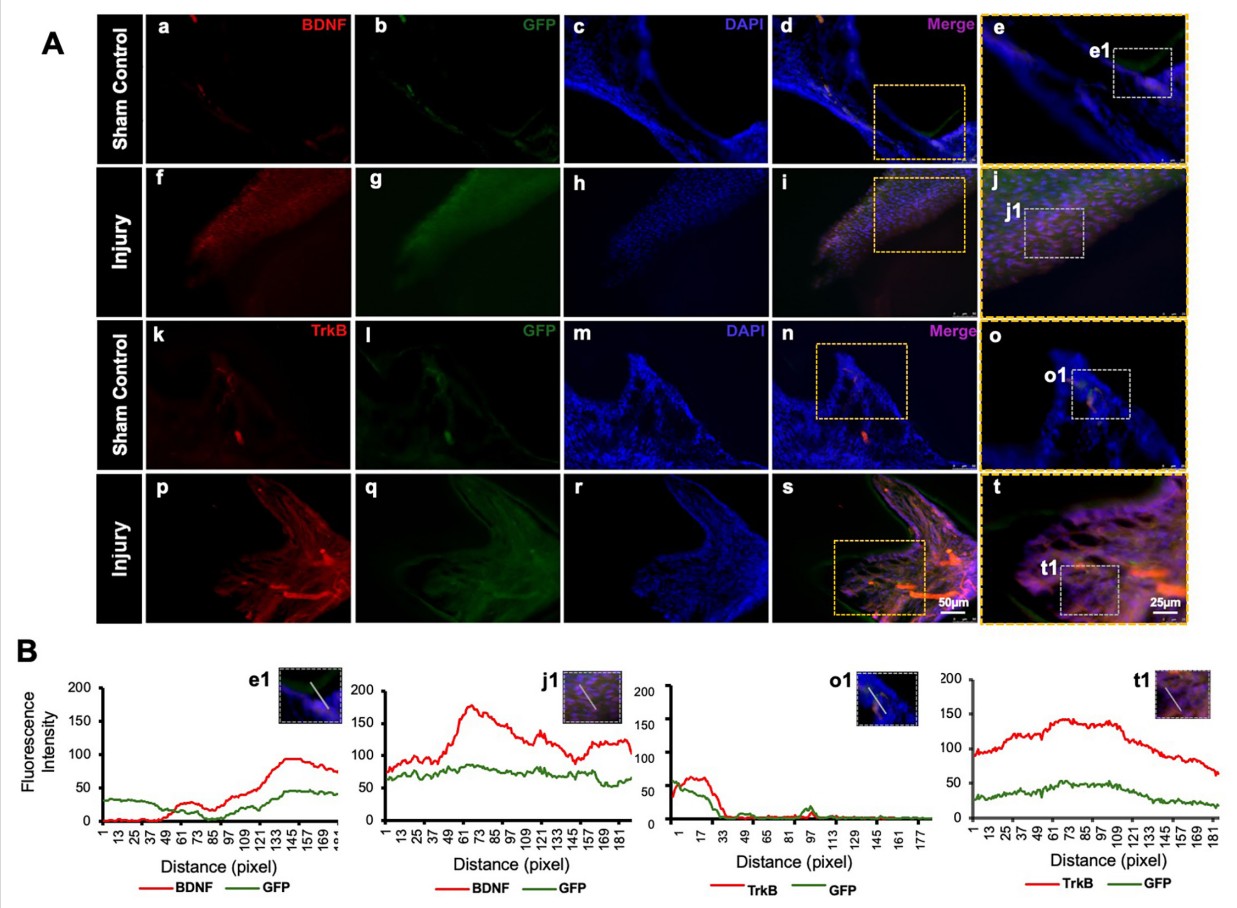

**Figure 6.** Detection of brain-derived neurotrophic factor (BDNF) and tropomyosin receptor kinase B (TrkB) in GFP-tagged cells transplanted in the pulp-capping mouse model. (**A**) Representative image of sham control of the pulp-capping mouse model. (**a–j**) Sham control vs. (**k–t**) representative image of the transplantation of Clustered Regularly Interspaced Short Palindromic Repeats (CRISPR)-engineered BDNF-overexpressing DPSCs in the pulp-capping mouse model. Scale bars: 25 and 50 µm. (**B**) Line graphs showing the co-localization of BDNF and GFP [**B** (**e1** and **j1**)] and of TrkB and GFP [**B** (**o1** and **t1**)].

groups. The sham control group (*Figure 5Aa–h*) exhibited a clear demarcation of the injured dentin area, highlighted by the white arrow indicating the drilling operation and the dotted line specifying the damaged region. In contrast, the experimental group treated with CRISPR-engineered BDNF-overexpressing DPSCs (*Figure 5Ai–p*) showed a notable increase in dentin formation at the injury site. A quantitative analysis of dentin density (*Figure 5D*) revealed a significant increase in the dentin volume in the group treated with CRISPR-engineered BDNF-overexpressing DPSCs compared to the sham control group. Specifically, the dentin volume in the treated group (*Figure 5C*) was 1315±42 mg HA/ccm, significantly higher than the control group (*Figure 5B*), which was measured at 1127±20.4 mg HA/ccm ($p<0.05$).

To evaluate morphological changes after transplantation, H&E staining was employed to assess cellular and structural changes in the dentin and pulp tissue following the transplantation of CRISPR-engineered BDNF-overexpressing DPSCs in the pulp-capping mouse model. *Figure 5* displays representative images of the sham control group and surgery group. The overall morphology of the tooth structure (*Figure 4D*) and higher magnifications of the injured area highlighted by the box are provided (*Figure 5*). In contrast to the control group, the BDNF-DPSC transplantation group demonstrated significant tooth regeneration and repair. The newly formed dentin at the injury site appears more organized and closely resembles the native dentin structure.

GFP staining allowed us to identify and localize the transfected areas within the injury site (*Figures 5 and 6*). Specifically, GFP expression was observed in areas correlating with the injury compared to the control (*Figure 6A*). This confirmed the successful transfection and localization of

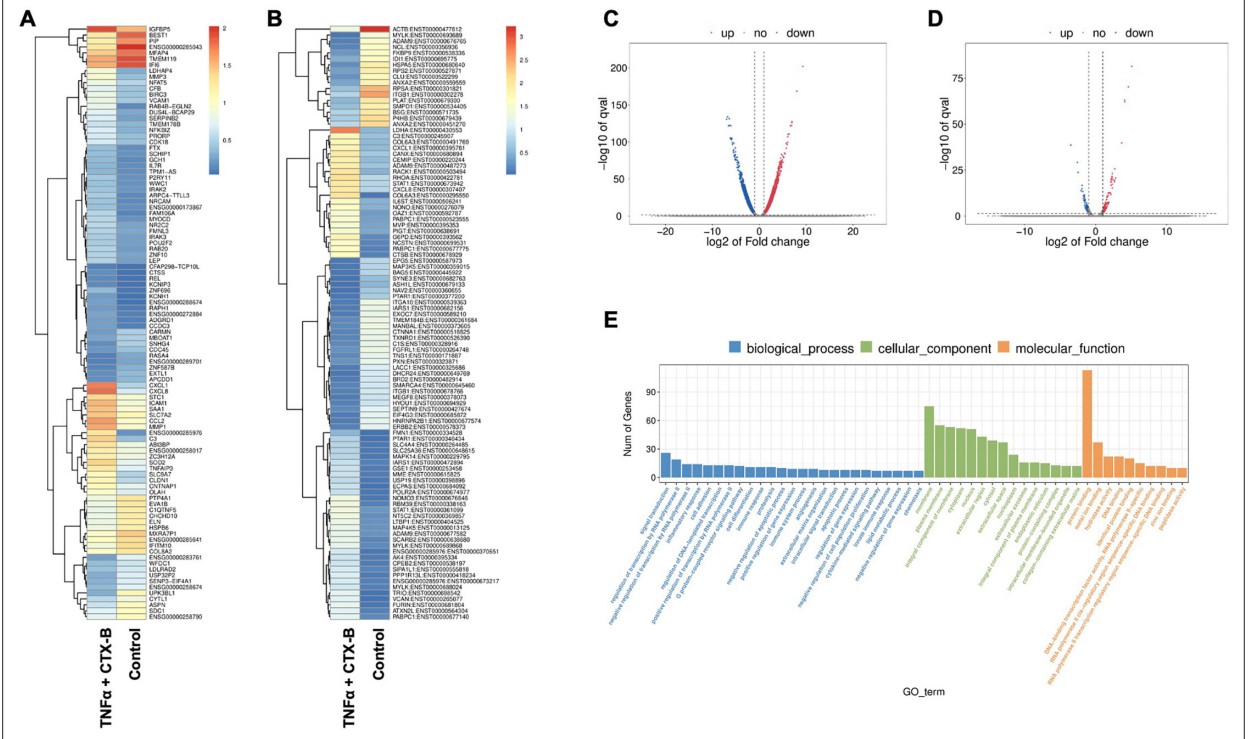

**Figure 7.** The total RNA-seq analysis between normal dental pulp stem cells (DPSCs) and tumor necrosis factor alpha (TNFα)+cyclotraxin-B (CTX-B)-treated DPSCs in dentinogenic media. (**A**) Heatmap of the differentially expressed genes between normal DPSCs and TNFα+CTX-B DPSCs in dentinogenic media from total RNA-seq analysis. (**B**) Transcripts heatmap of the differentially expressed genes between normal DPSCs and TNFα+CTX-B DPSCs in dentinogenic media from total RNA-seq analysis. Color corresponds to log to fold change. Red and yellow stripes in the figure represent high-expression genes, while blue stripes represent low-expression genes. (**C**) Volcano map comparing gene expression between normal DPSCs and TNFα+CTX-B-treated DPSCs. (**D**) Volcano map comparing transcripts between normal DPSCs and TNFα+CTX-B-treated DPSCs. In both plots, the x-axis represents the log2 fold change in gene expression, and the y-axis represents the -log10 of the adjusted p-value. Genes or transcripts significantly upregulated are marked in red, and downregulated genes are marked in blue. The dotted horizontal line indicates the threshold for statistical significance, while vertical lines indicate fold change thresholds. (**E**) Significant enriched Gene Ontology (GO) terms among control and TNFα+CTX-B-treated DPSCs based on biological process (blue), cellular function (green), and molecular function (orange). Bar chart representing the number of genes associated with significant GO terms. The x-axis lists the GO terms, while the y-axis shows the number of genes enriched for each term.

the CRISPR-engineered DPSCs in the mandible of the mouse tooth. Line graphs show colocalizations of GFP-tagged DPSCs expressing BDNF and TrkB (*Figure 6B*). These results indicate that DPSCs overexpressing BDNF, engineered using CRISPR and transplanted into a pulp-capping mouse model, significantly improve dentin regeneration.

## RNA sequencing of TNFα-induced DPSCs odontogenic differentiation mediated by TrkB

In our investigation of the combination of TNFα and cyclotraxin-B (CTX-B, a TrkB antagonist) treatment compared to the control, we further aimed to identify the transcriptomic differences associated with the involvement of the BDNF-TrkB pathway related to inflammation. The gene heatmap and transcripts heatmap revealed distinct high and low gene expression patterns between the control and TNFα+CTX-B groups (*Figure 7A and B*). The volcano plot of differentially expressed genes (DEGs) illustrated the significant downregulation and upregulation in gene expression between the control and TNFα+CTX-B groups (*Figure 7C and D*).

We identified significantly enriched Gene Ontology (GO) terms in the categories of biological process, cellular component, and molecular function between these groups (*Figure 7E*). Notably, there was a distinct difference in membrane-related cellular functions and molecular functions related to protein binding. The GO enrichment pathway analysis revealed that the highest p-values were

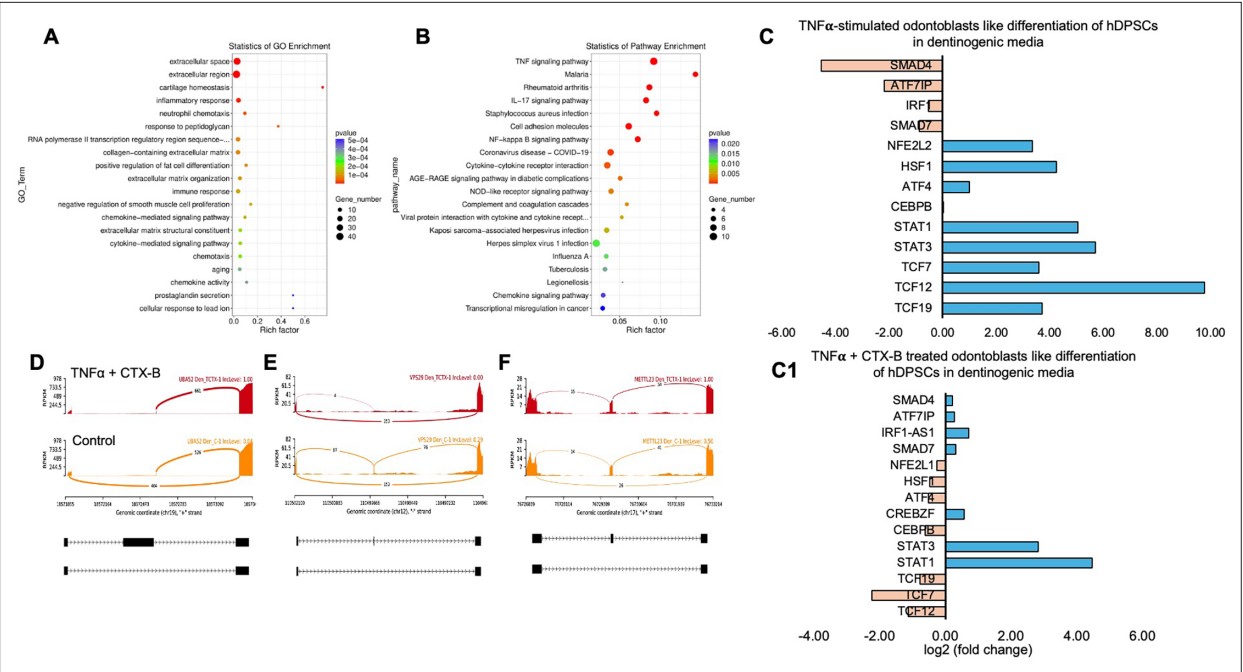

**Figure 8.** Signaling pathways affected by tumor necrosis factor alpha (TNFα)-induced dental pulp stem cells (DPSCs) mediated by tropomyosin receptor kinase B (TrkB) inhibitor in odontoblasts like differentiation of DPSCs in dentinogenic media. (**A**) A dot plot showing the Gene Ontology (GO) terms enriched in the dataset. The size of the dots represents the number of differentially expressed genes (DEGs), while the color indicates the p-value, with darker colors representing more significant enrichment depending on the rich factor, which is the ratio of the number of DEGs annotated in a GO term to the total number of genes annotated in that term. (**B**) The pathways enriched among the DEGs. The x-axis shows the rich factor, and the y-axis lists the pathways. The size of the dot indicates the number of genes involved in each pathway. (**C–C1**) Next-generation RNA sequencing was done using the poly-A-RNA sequencing technique. Histogram showing upregulated and activated transcription factors (blue) and repressed or downregulated transcription factors (orange). Histogram showing the effect of TNFα alone and combined with cyclotraxin-B (CTX-B) on the regulation of various transcription factors. Notably, *TCF (7, 12, 19)* was abolished with the treatment of CTX-B. (**D–F**) The Sashimi plots show the expression patterns of specific genes in response to TNFα+CTX-B treatment. Data were examined on sashimi plots, which revealed the number of variants and genomic mutations on chr19, chr12, and chr17 in TNFα+CTX-B-treated cells in dentinogenic media against control. Red sashimi plots show variants in the TNFα+CTX-B-treated group, and orange shows in the control. The lower black annotations are Read alignments of alternative isoforms and genomic regions of interest providing the gene model, with exons represented as thick blocks and introns as thin lines.

associated with the extracellular space and region (*Figure 8A*). Furthermore, Reactome pathway analysis indicated high p-values for pathways related to TNF signaling (*Figure 8B*).

Sashimi plots comparing control and TNFα+CTX-B-treated DPSCs highlighted differentially spliced exons in genomic regions of interest (*Figure 8D–F*). The transcriptional response of TNFα+CTX-B-treated odontoblast-like differentiated DPSCs showed significant downregulation in TCFs and *ATF4* and upregulation in signal transducers and activators of *STAT1* and *STAT3* (*Figure 8C1*) compared to the TNFα alone-treated group (*Figure 8C*). These findings suggest that TNFα and CTX-B treatment induces significant transcriptomic alterations, particularly affecting pathways involved in cellular signaling and extracellular matrix interactions.

## Discussion

Odontoblastic differentiation of DPSCs in response to caries injury takes place within an inflammatory environment (*Chmilewsky et al., 2016*). However, there is still limited information on the role of inflammation in reparative dentinogenesis and the biology of DPSCs. Our findings demonstrate the activation of TrkB during the caries process. Consistent with this, our in vitro data confirmed that various inflammatory mediators increase TrkB expression and activation in DPSCs. These findings suggest that BDNF/TrkB directly affects the pulpal response to caries. In vivo, TrkB is more highly expressed in inflamed pulp compared to normal tissue. The transplantation of CRISPR-engineered BDNF-overexpressing DPSCs in a pulp-capping mouse model significantly enhances dentin regeneration

and repair, as demonstrated by increased dentin volume and improved tissue morphology. Furthermore, RNA sequencing reveals significant transcriptomic changes in TNFα-treated DPSCs, affecting cellular signaling and extracellular matrix pathways. These results underscore the critical role of TrkB in inflammatory responses and dentin repair, suggesting potential for targeted TrkB modulation in dental tissue regeneration.

TNFα and LTA are known to mediate inflammatory responses and contribute to the recruitment and activation of immune cells at the injury site (*Farges et al., 2015*; *Kawai and Akira, 2010*). This inflammatory response is crucial for eliminating pathogens and debris, setting the stage for tissue repair. In the context of dentin formation, these cytokines promote the differentiation and activity of odontoblast-like cells, which are responsible for producing the reparative dentin matrix. (*Durand et al., 2006*; *Keller et al., 2010*) LPS, components of the outer membrane of Gram-negative bacteria, also stimulate an inflammatory response when they infiltrate the dental pulp (*Brodzikowska et al., 2022*). The presence of LPS triggers the release of various pro-inflammatory cytokines, further enhancing the recruitment of immune cells and promoting the processes necessary for tertiary dentin formation (*Widbiller et al., 2018*). These pro-inflammatory cytokines coordinate a complex cascade of events that lead to the regeneration of dentin, highlighting their importance in dental tissue repair mechanisms.

Our study confirms the impact of pro-inflammatory stimuli on the expression and activation of TrkB in DPSCs. Immunofluorescence analysis (*Figure 1*) demonstrated that exposure to LPS, LTA, and TNFα significantly upregulated TrkB and its phosphorylated form, p-TrkB, compared to the control. This suggests these inflammatory agents can activate TrkB signaling pathways in DPSCs, potentially influencing their behavior and differentiation. TNFα showed the most pronounced effect, indicating its potent role in activating TrkB signaling. These findings are consistent with previous studies indicating that TNFα can enhance TrkB expression and activation in various cell types, promoting cellular responses to inflammation (*Irfan et al., 2024*; *Kim et al., 2023b*; *Paula-Silva et al., 2009*). These results suggest that pro-inflammatory stimuli such as LPS, LTA, and TNFα can significantly enhance TrkB and p-TrkB expression in DPSCs.

Furthermore, our data demonstrate significant upregulation of TrkA and TrkB expressions in dentinogenic media in response to inflammatory components. TrkB expression was notably increased by LPS, LTA, TNFα, and C5a, suggesting widespread receptor activation. This upregulation may play a critical role in mediating the cellular responses to inflammation, potentially influencing the behavior of DPSCs in the context of dental pulp inflammation and regeneration. Particularly, TNFα induced the highest TrkB upregulation in both media types, emphasizing the role of TrkB in mediating responses to neurotrophic factors and involvement in inflammatory pathways (*Irfan et al., 2022a*). The differential expression in dentinogenic media indicates enhanced receptor sensitivity and signaling in an inflammatory environment (*Zhang et al., 2016*). A dose-dependent increase in TrkB expression was observed with escalating TNFα concentrations, with significant upregulation at 10 ng/mL, beyond which further increases in TNFα did not enhance expression. This suggests that higher doses are unnecessary to avoid inflammation related to degeneration or infection (*Kim et al., 2023b*). The time-dependent response to a fixed TNFα concentration showed a peak at day 7, indicating that a minimum of 7 days of differentiation is necessary for culturing DPSCs in inflammatory responses related to TrkB expression. In vivo, TrkB expression was significantly higher in the inflamed pulp tissue of carious teeth, indicating that TrkB may play a critical role in the response to dental caries and pulp inflammation. This suggests that BDNF-TrkB signaling could be crucial in the neurogenic and angiogenic processes involved in pulp tissue repair (*Bar et al., 2021*; *de Moraes et al., 2018*).

The transplantation of CRISPR-engineered BDNF-overexpressing DPSCs has demonstrated promising results in enhancing dentin regeneration in a pulp-capping mouse model. Micro-CT images and H&E staining analyses indicate that the overexpression of BDNF via CRISPR engineering significantly promotes dentin repair at the injury site. BDNF is known for its role in molecular and physiological involvement (*Lu et al., 2014*). Its application in dental tissue engineering suggests it may also play a crucial role in odontogenesis and dentinogenesis (*Kim et al., 2023b*). The significant increase in dentin volume in the micro-CT experiment observed in this study supports the hypothesis that BDNF can enhance the regenerative capabilities of DPSCs, leading to improved outcomes in dentin repair. Enhanced dentinogenesis was evidenced by the well-organized and structurally similar new dentin formation in the transplantation group compared to the native dentin structure. The histological

analysis with H&E staining further verified these findings, showing a marked improvement in dentin formation and reduced inflammation in the BDNF-DPSC transplantation group compared to the control group. This suggests that BDNF enhances dentin regeneration, creating a conducive environment for tissue repair. Several potential mechanisms could explain that BDNF may enhance the proliferation and differentiation of DPSCs into odontoblast-like cells, promoting reparative dentin formation (*Tsutsui, 2020*). Moreover, the neurotrophic properties of BDNF could support the survival and function of sensory nerve fibers within the pulp, which may play a role in regulating the repair process (*Bathina and Das, 2015*).

In vitro experiments with DPSCs further supported these findings, demonstrating that TrkB expression is significantly upregulated in response to TNFα treatment. This suggests that inflammation enhances the neurogenic potential of DPSCs through TrkB signaling. The co-expression of TrkB in

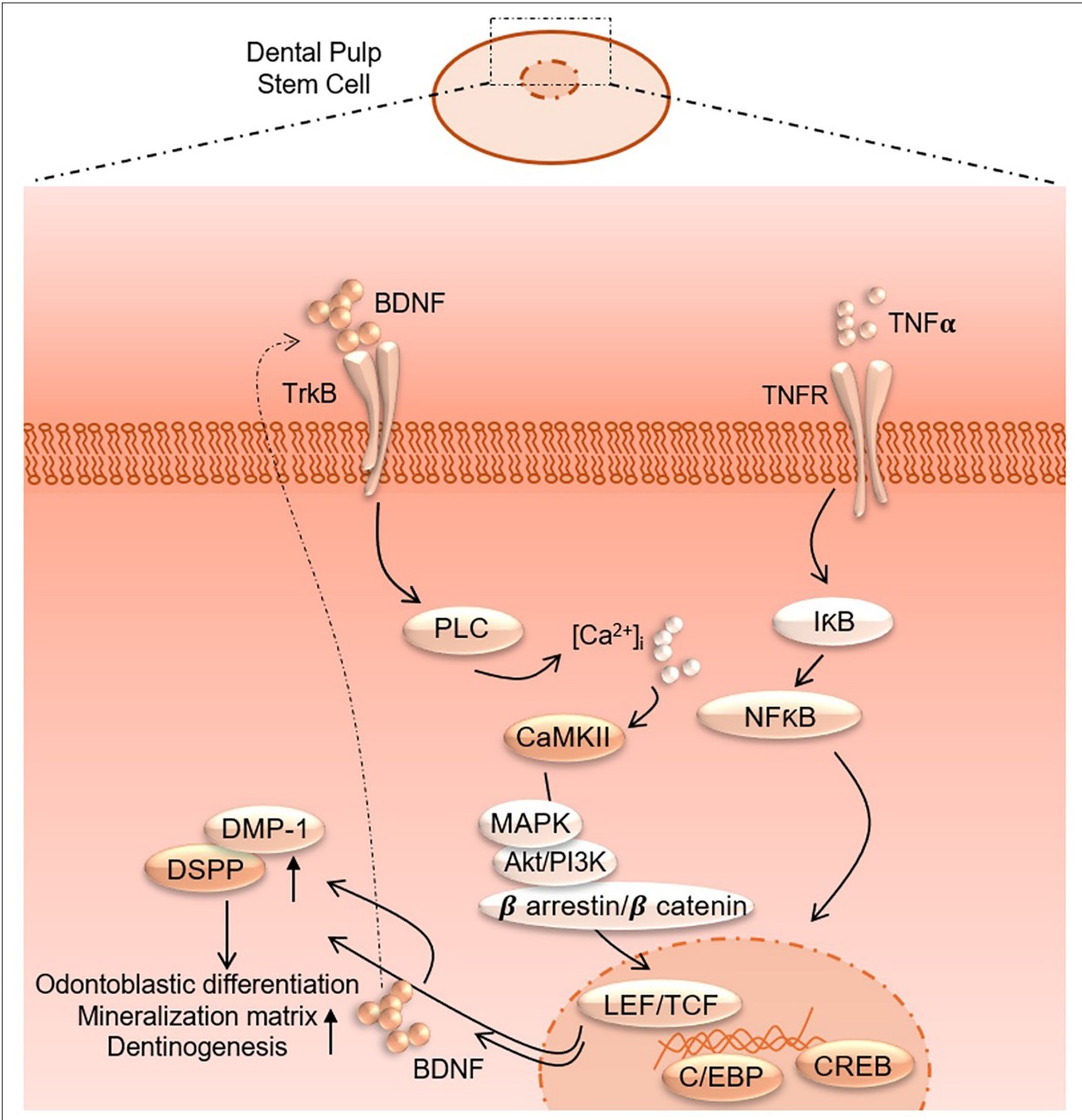

**Figure 9.** Proposed interaction of tumor necrosis factor alpha (TNFα) and brain-derived neurotrophic factor (BDNF)/tropomyosin receptor kinase B (TrkB) downstream signaling to modulate odontoblastic differentiation in dental pulp stem cells (DPSCs).

both carious teeth and DPSCs highlights the potential of targeting the BDNF-TrkB pathway in dental regenerative therapies. Modulating the inflammatory environment to enhance TrkB signaling could improve the regenerative properties of DPSCs, offering promising strategies for managing dental caries and pulpitis. The combined treatment of TNFα and CTX-B elucidates the significant transcriptional reprogramming compared with the control. The volcano plots highlight a substantial number of DEGs, indicating upregulation and downregulation of gene expression compared to control. The GO enrichment analysis reveals that these DEGs are predominantly associated with biological processes such as extracellular matrix organization, inflammatory response, and cytokine-mediated signaling pathways. This is further supported by the GO term statistics, where notable terms like extracellular space and cartilage homeostasis are enriched, suggesting alterations in the cellular microenvironment and immune response.

Additionally, the pathway enrichment analysis identifies critical pathways such as TNF signaling, *NF-κB* signaling, and cytokine–cytokine receptor interaction, underscoring the crucial role of inflammatory signaling in the observed transcriptional changes. Finally, the gene expression analysis under TNFα+CTX-B treatment provides specific examples of genes with altered expression patterns, confirming the extensive impact of the treatment at the molecular level. These findings suggest that TNFα treatment mediated by the BDNF/TrkB pathway induces a complex network of transcriptional changes that modulate the extracellular matrix and inflammatory signaling pathways (*Figure 9*). We believe that several transcriptional factors are involved in BDNF/TrkB downstream signaling, especially the *TCF* family. Further studies are required to validate this interaction, leading to a smooth line between the stem cell therapy and engineering era. Also, it is necessary to investigate the long-term effects of BDNF overexpression on dentin regeneration and to explore the underlying mechanisms driving this enhanced repair process. Additionally, assessing the potential for clinical translation of this approach will be crucial in determining its viability as a therapeutic option for dental pulp injuries and related conditions.

The use of CRISPR-engineered BDNF-overexpressing DPSCs could offer a novel and effective approach for enhancing dentin repair and regeneration in cases of dental pulp injury and significant clinical implications for dental regenerative therapies. This approach could potentially reduce the need for more invasive treatments, such as root canal therapy, and improve outcomes for patients with dental trauma or disease.

## Acknowledgements

This study was supported by the NIH/NIDCR Grant: R01 DE029816-SC.

## Additional information

### Funding

| Funder | Grant reference number | Author |
|---|---|---|
| National Institute of Dental and Craniofacial Research | R01 DE029816 | Seung Chung |

The funders had no role in study design, data collection and interpretation, or the decision to submit the work for publication.

### Author contributions

Ji Hyun Kim, Software, Validation, Investigation, Visualization, Methodology, Writing – original draft; Muhammad Irfan, Data curation, Formal analysis, Validation, Investigation, Visualization, Methodology, Writing – original draft; Sreelekshmi Sreekumar, Software, Validation, Investigation, Methodology; Atsawasuwan Phimon, Stephanie Kim, Validation, Investigation, Methodology; Seung Chung, Conceptualization, Data curation, Software, Funding acquisition, Validation, Investigation, Methodology, Writing – original draft, Project administration, Writing - review and editing

### Author ORCIDs

Muhammad Irfan (iD) https://orcid.org/0000-0002-9788-8231

Seung Chung https://orcid.org/0000-0003-1449-8762

### Ethics

The Institutional Animal Care and Use Committee (IACUC) at the University of Illinois Chicago reviewed and approved all surgical and experimental procedures. C57BL/6 male mice, 6-8 weeks old (n = 12), were used for the experiments and purchased from Jackson Laboratory (#000461). For the mouse molars, the mandibles of the C57BL/6 mice were collected for further experimentation (Protocol no.: 21-197).

Joint Public Review: https://doi.org/10.7554/eLife.105153.3.sa1
Author response https://doi.org/10.7554/eLife.105153.3.sa2

---

## Additional files

### Supplementary files

MDAR checklist

### Data availability

RNA gene profiling data, raw IHC images and statical data have been deposited to https://doi.org/10.5061/dryad.c866t1gkc.

The following dataset was generated:

| Author(s) | Year | Dataset title | Dataset URL | Database and Identifier |
|---|---|---|---|---|
| Kim J, Irfan M, Sreekumar S, Phimon A, Kim S, Chung S | 2025 | CRISPR-edited DPSCs, constitutively expressing BDNF enhance dentin regeneration in injured teeth | https://doi.org/10.5061/dryad.c866t1gkc | Dryad Digital Repository, 10.5061/dryad.c866t1gkc |

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
