## [Editor Report · eLife Assessment]

This study on the effect of the trophic factor BDNF upon dental cells is an understudied subject that is relevant to dental regeneration and repair. Given that the topic is new and has not been covered previously, the report is a **useful** foray into a new area of investigation, although several experimental results could be strengthened. The connection of BDNF and dental health is a **solid** attempt in potentially translating trophic factor signaling clinically, which has been stymied in past efforts.

---

## [Referee Report · Joint Public Review]

This work employs both in vitro and in vivo methods to investigate the contribution of BDNF/TrkB signaling to enhancing differentiation and dentin-repair capabilities of dental pulp stem cells in the context of exposure to a variety of inflammatory cytokines. A particular emphasis of the approach is employment of dental pulp stem cells in which BDNF expression has been enhanced using CRISPR technology. Transplantation of such cells are proposed to improve dentin regeneration in a mouse model of tooth decay. The study provides several interesting findings, including demonstrating that exposure to several cytokines/inflammatory agents increases the quantity of activated phospho-Trk B in dental pulp stem. One issue that was not covered is the involvement of the p75 neurotrophin receptor which is also highly sensitive to inflammation and injury. The conclusions could be further augmented by demonstrating the specificity of the antibodies via immunoblot methods, both in the presence and absence of BDNF and other neurotrophins, NT-3 and NT-4, which can also bind to the TrkB receptor.

---

## [Author Response]

The following is the authors’ response to the original reviews

**Public Reviews:**

**Reviewer #1 (Public review):**
This work employs both in vitro and in vivo/transplant methods to investigate the contribution of BDNF/TrkB signaling to enhancing differentiation and dentin-repair capabilities of dental pulp stem cells in the context of exposure to a variety of inflammatory cytokines. A particular emphasis of the approach is the employment of dental pulp stem cells in which BDNF expression has been enhanced using CRISPR technology. Transplantation of such cells is said to improve dentin regeneration in a mouse model of tooth decay.The study provides several interesting findings, including demonstrating that exposure to several cytokines/inflammatory agents increases the quantity of (activated) phospho-Trk B in dental pulp stem cells.However, a variety of technical issues weaken support for the major conclusions offered by the authors. These technical issues include the following:

Thank you for your keen observation and evaluation, which helped us significantly improve our manuscript. We have addressed the concerns and comments point by point in detail and substantially revised the manuscript and Figures. We hope that the manuscript is acceptable in the current improvised version.

Detailed response to your comments/concerns is as follows:

(1) It remains unclear exactly how the cytokines tested affect BDNF/TrkB signaling. For example, in Figure 1C, TNF-alpha increases TrkB and phospho-TrkB immunoreactivity to the same degree, suggesting that the cytokine promotes TrkB abundance without stimulating pathways that activate TrkB, whereas in Figure 2D, TNF-alpha has little effect on the abundance of TrkB, while increasing phospho-TrkB, suggesting that it affects TrkB activation and not TrkB abundance.

Thank you for your kind concern. Recently, we have demonstrated the effect and interaction of TNF-alpha and Ca2+/calmodulin-dependent protein kinase II on the regulation of the inflammatory hDPSCs dentino-differentiation via BDNF/TrkB receptor signaling using TrkB inhibitor (Ref. below, and Figure 9). Moreover, we agree with your concern, and we have re-analyzed our replicates and found a better trend and significant abundance of TrkB as well (please refer to revised Figure 2D).

Ref.: Kim, Ji Hyun, et al. (2025) "Ca 2+/calmodulin-dependent protein kinase II regulates the inflammatory hDPSCs dentino-differentiation via BDNF/TrkB receptor signaling." Frontiers in Cell and Developmental Biology 13: 1558736.

(2) I find the histological images in Figure 3 to be difficult to interpret. I would have imagined that DAPI nuclear stains would reveal the odontoblast layer, but this is not apparent. An adjacent section labeled with conventional histological stains would be helpful here. Others have described Stro-1 as a stem cell marker that is expressed on a minority of cells associated with vasculature in the dental pulp, but in the images in Figure 3, Stro-l label is essentially co-distributed with DAPI, in both control and injured teeth, indicating that it is expressed in nearly all cells. Although the authors state that the Stro-1-positive cells are associated with vasculature, but I see no evidence that is true.

Thank you for your concern. STRO-1 is a mesenchymal stem cell marker also expressed in dental pulp stem cells; both populations are distributed in the pulp. DPSCs can contribute to tissue repair and regeneration in inflamed pulp by differentiating into odontoblasts and forming reparative dentin. Moreover, in the case of carious and inflamed pulp, they are disorganized depending on the extent of infection/injury. Our purpose here was to point out DPSCs presence, not vasculature, which will differentiate into odontoblasts in such a scenario. We have revised Figure 3 by adding magnified images and dotted lines to indicate the boundary between the pulp and dentin.

Ref. Volponi A. A., Pang Y., Sharpe P. T. Stem cell-based biological tooth repair and regeneration. Trends in Cell Biology. 2010;20(12):715–722.

(3) The data presented convincingly demonstrate that they have elevated BDNF expression in their dental pulp stem cells using a CRISPR-based approach I have a number of questions about these findings. Firstly, nowhere in the paper do they describe the nature of the CRISPR plasmid they are transiently transfecting. Some published methods delete segments of the BDNF 3'-UTR while others use an inactivated Cas9 to position an active transactivator to sequences in the BDNF promoter. If it is the latter approach, transient transfection will yield transient increases in BDNF expression. Also, as BDNF employs multiple promoters, it would be helpful to know which promoter sequence is targeted, and finally, knowing the identity of the guide RNAs would allow assessment for the potential of off-target effects I am guessing that the investigators employ a commercially obtained system from Santa Cruz, but nowhere is this mentioned. Please provide this information.

Dear Reviewer, yes, you are right. We have used a commercially obtained system from Santa Cruz, i.e., BDNF CRISPR Activation Plasmid (h): sc-400029-ACT and UltraCruz Transfection Reagent (sc-395739), and they have been mentioned in Chemicals and Reagents section of Materials and Methods as follows.

“BDNF CRISPR Activation Plasmid (h) is a synergistic activation mediator (SAM) transcription activation system designed to upregulate gene expression specifically BDNF CRISPR Activation Plasmid (h) consists of three plasmids at a 1:1:1 mass ratio: a plasmid encoding the deactivated Cas9 (dCas9) nuclease (D10A and N863A) fused to the transactivation domain VP64, and a blasticidin resistance gene; a plasmid encoding the MS2-p65-HSF1 fusion protein, and a hygromycin resistance gene; a plasmid encoding a target-specific 20 nt guide RNA fused to two MS2 RNA aptamers, and a puromycin resistance gene.”

The resulting SAM complex binds to a site-specific region approximately 200-250 nt upstream of the transcriptional start site and provides robust recruitment of transcription factors for highly efficient gene activation

Following transfection, gene activation efficiency could be assayed by WB, IF, or IHC using antibody: pro-BDNF Antibody (5H8): sc-65514

**Author response image 1. sa2fig1:** 

(4) Another question left unresolved is whether their approach elevated BDNF, proBDNF, or both. Their 28 kDa western blot band apparently represents proBDNF exclusively, with no mature BDNF apparent, yet only mature BDNF effectively activates TrkB receptors. On the other hand, proBDNF preferentially activates p75NTR receptors. The present paper never mentions p75NTR, which is a significant omission, since other investigators have demonstrated that p75NTR controls odontoblast differentiation.

Dear reviewer, thank you for your noticing the error.

Pro-BDNF is produced as a 32-kDa precursor that undergoes N-glycosylation and glycosulfation on residues located within the pro-domain of the precursor. N-terminal cleavage of the precursor generates mature BDNF as well as a minor truncated form of the precursor (28 kDa) that arises by a different processing mechanism than mature BDNF. The precursor undergoes N-terminal cleavage within the trans-Golgi network and/or immature secretory vesicles to generate mature BDNF (14 kDa).

We checked our data and band size, and it shows a little mistake (Thank you for your keen observation and pointing out). The CRISPR protocol required verification of gene activation by checking pro-BDNF, as mentioned in the methodology. The labeling has been revised in the figure as pro-BDNF, and the actual blot with a ladder has been shown below for clarification.

(5) In any case, no evidence is presented to support the conclusion that the artificially elevated BDNF expression has any effect on the capability of the dental pulp stem cells to promote dentin regeneration. The results shown in Figures 4 and 5 compare dentin regeneration with BDNF-over-expressing stem cells with results lacking any stem cell transplantation. A suitable control is required to allow any conclusion about the benefit of over-expressing BDNF.

We have tested the presence of BDNF overexpressing cells by the higher expression of GFP here. Moreover, a significant increment in the dentin mineralization volume indicates the advantage of BDNF-over-expressing stem cells. Recently, we published the in vitro effects of BDNF/TrkB on DPSCs odontoblastic differentiation strongly supporting our in vivo data. Currently, we are in a difficult position to conduct the animal study within a short period of time. We would definitely consider using positive control in our future studies.

Ref.: Kim, Ji Hyun, et al. (2025) "Ca 2+/calmodulin-dependent protein kinase II regulates the inflammatory hDPSCs dentino-differentiation via BDNF/TrkB receptor signaling." Frontiers in Cell and Developmental Biology 13: 1558736.

(6) Whether increased BDNF expression is beneficial or not, the evidence that the BDNF-overexpressing dental pulp stem cells promote dentin regeneration is somewhat weak. The data presented indicate that the cells increase dentin density by only 6%. The text and figure legend disagree on whether the p-value for this effect is 0.05 or 0.01. In either case, nowhere is the value of N for this statistic mentioned, leaving uncertainty about whether the effect is real.

A significant increment in the dentin mineralization volume by about 7.76% indicates the advantage of BDNF-over-expressing stem cells, and we believe this could be a breakthrough to advance stem cell engineering and therapy further to get this percentage higher in the future. The text in the result section shows that the p-value for this effect is 0.05. While N was 3 previously, we analyzed two more samples by CT scan and revised results, taking N = 5, which improved the results a little more to about 8.53%. Thank you for noticing; the figure legend has been corrected to 0.05.

Similarly, our in vitro data in the current study supports the notion that it adds up to mineralization and odontoblastic differentiation. We recently published that BDNF/TrkB significantly enhances calcium deposits and mineralization using a battery of in vitro experiments.

Ref.: Kim, Ji Hyun, et al. (2025) "Ca 2+/calmodulin-dependent protein kinase II regulates the inflammatory hDPSCs dentino-differentiation via BDNF/TrkB receptor signaling." Frontiers in Cell and Developmental Biology 13: 1558736.

(7) The final set of experiments applies transcriptomic analysis to address the mechanisms mediating function differences in dental pulp stem cell behavior. Unfortunately, while the Abstract indicates " we conducted transcriptomic profiling of TNFα-treated DPSCs, both with and without TrkB antagonist CTX-B" that does not describe the experiment described, which compared the transcriptome of control cells with cells simultaneously exposed to TNF-alpha and CTX-B. Since CTX-B blocks the functional response of cells to TNF-alpha, I don't understand how any useful interpretation can be attached to the data without controls for the effect of TNF alone and CTX-B alone.

Dear reviewer, yes, we did it alone and together as well. Earlier, we showed only the combined results and mentioned the interaction between TNFα and TrkB. We have included the results from TNFα alone and combined them with CTX-B for better comparison (Please refer to Figure 8). Figure 8C1 clearly shows the reversal of certain factors with the treatment of TrkB inhibitor compared to figure 8C with TNFα alone treated group.

**Reviewer #2 (Public review):**
Summary:In this manuscript, the authors investigate the potential for overexpressing BDNF in dental pulp stem cells to enhance dentin regeneration. They suggest that in the inflammatory environment of injured teeth, there is increased signaling of TrkB in response to elevated levels of inflammatory molecules.Strengths:The potential application to dentin regeneration is interesting.Weaknesses:There are a number of concerns with this manuscript to be addressed.

Thank you for your compliments, keen observation, and evaluation, which helped us significantly improve our manuscript. We have addressed the concerns and comments point by point in detail and substantially revised the manuscript and Figures. We hope that the manuscript is acceptable in the current improvised version.

Detailed response to your comments/concerns is as follows:

(1) Insufficient citation of the literature. There is a vast literature on BDNF-TrkB regulating survival, development, and function of neurons, yet there is only one citation (Zhang et al 2012) which is on Alzheimer's disease.

More references have been cited accordingly.

(2) There are several incorrect statements. For example, in the introduction (line 80) TrkA is not a BDNF receptor.

Thank you for noticing the typo; the sentence has been corrected.

(3) Most important - Specific antibodies must be identified by their RRID numbers. To state that "Various antibodies were procured:... from BioLegend" is unacceptable, and calls into question the entire analysis. Specifically, their Western blot in Figure 4B indicates a band at 28 kDa that they say is BDNF, however the size of BDNF is 14 kDa, and the size of proBDNF is 32 and 37 kDa, therefore it is not clear what they are indicating at 28 kDa. The validation is critical to their analysis of BDNF-expressing cells.

Dear reviewer, thank you for your kind concern. Sorry for the inconvenience; we have added RRID numbers of antibodies.

Pro-BDNF is produced as a 32-kDa precursor that undergoes N-glycosylation and glycosulfation on residues located within the pro-domain of the precursor. N-terminal cleavage of the precursor generates mature BDNF as well as a minor truncated form of the precursor (28 kDa) that arises by a different processing mechanism than mature BDNF. The precursor undergoes N-terminal cleavage within the trans-Golgi network and/or immature secretory vesicles to generate mature BDNF (14 kDa).

We checked our data and band size, and it shows a mistake in recognizing ladder size. It is actually a 14kDa band which has been shown. The labeling has been revised in the figure, and the actual blot with a ladder has been shown below for clarification. Similarly, our data focused on the fact that the observed cellular effects are more consistent with BDNF/TrkB-mediated pathways, which are known to promote survival and differentiation.

(4) Figure 2 indicates increased expression of TrkB and TrkA, as well as their phosphorylated forms in response to inflammatory stimuli. Do these treatments elicit increased secretion of the ligands for these receptors, BDNF and NGF, respectively, to activate their phosphorylation? Or are they suggesting that the inflammatory molecules directly activate the Trk receptors? If so, further validation is necessary to demonstrate that.

Thank you for your kind concern. TNF-α increases the number of TrkB receptors. The enhanced TrkB activation may result from a greater number of receptors and/or increased activation of individual receptors. In either case, inflammatory agents enhance the TrkB receptor signaling pathway.

Recently, we have demonstrated the effect and interaction of TNF-alpha and Ca2+/calmodulin-dependent protein kinase II on the regulation of the inflammatory hDPSCs dentino-differentiation via BDNF/TrkB receptor signaling using TrkB inhibitor (Ref. below, and Figure 9). For now, we have added figure 9 for the proposed mechanism of action based on our recent and current study.

Ref.: Kim, Ji Hyun, et al. (2025) "Ca 2+/calmodulin-dependent protein kinase II regulates the inflammatory hDPSCs dentino-differentiation via BDNF/TrkB receptor signaling." Frontiers in Cell and Developmental Biology 13: 1558736.

(5) Figure 7 - RNA-Seq data, what is the rationale for treatment with TNF+ CTX-B? How does this identify any role for TrkB signaling? They never define their abbreviations, but if CTX-B refers to cholera toxin subunit B, which is what it usually refers to, then it is certainly not a TrkB antagonist.

Thank you for your concern. Cyclotraxin-B (CTX-B) is a TrkB antagonist (mentioned in the revised manuscript). In order to identify the underlying mechanism, we ought to locate certain transcriptional factors interacting with the TrkB/BDNF signaling, leading to differentiation and dentinogenesis. Therefore, we treated it with a TrkB inhibitor.

Earlier, we showed only the combined results and mentioned the interaction between TNFα and TrkB. We have included the results from TNFα alone and combined them with CTX-B for better comparison (Please refer to Figure 8). Figure 8C1 clearly shows the reversal of certain factors with the treatment of TrkB inhibitor compared to figure 8C with TNFα alone treated group. We agree that the precise role of CTX-B in modulating TrkB signaling requires further clarification and have now included this point in the revised discussion while we are currently working on this aspect.

**Reviewer #3 (Public review):**
In general, although the authors interpret their results as pointing towards a possible role of BDNF in dentin regeneration, the results are over-interpreted due to the lack of proper controls and focus on TrkB expression, but not its isoforms in inflammatory processes. Surprisingly, the authors do not study the possible role of p75 in this process, which could be one of the mechanisms intervening under inflammatory conditions.

Thank you for your compliments, keen observation, and evaluation, which helped us significantly improve our manuscript. We have addressed the concerns and comments point by point in detail and substantially revised the manuscript and Figures. We hope that the manuscript is acceptable in the current improvised version.

Detailed response to your comments/concerns is as follows:

(1) The authors claim that there are two Trk receptors for BDNF, TrkA and TrkB. To date, I am unaware of any evidence that BDNF binds to TrkA to activate it. It is true that two receptors have been described in the literature, TrkB and p75 or NGFR, but the latter is not TrkA despite its name and capacity to bind NGF along with other neurotrophins. It is crucial for the authors to provide a reference stating that TrkA is a receptor for BDNF or, alternatively, to correct this paragraph.

Dear reviewer, we apologize for the inconvenience; it was an error. BDNF binds to TrkB, and the sentence has been corrected.

(2) The authors discuss BDNF/TrkB in inflammation. Is there any possibility of p75 involvement in this process?

Mature BDNF binds to the high-affinity receptor tyrosine kinase B (TrkB), activating signaling cascades, while pro-BDNF binds to the p75 neurotrophin receptor (p75NTR). So, we don’t think there’s a possibility, as our data shows mature BDNF production. Here, we initially screened the TrkA and TrkB involvement in dentinogenesis and chose to work with BDNF and its receptor TrkB. Future studies can be directed to elucidate its mechanism of action in the context of dentinogenesis.

(3) The authors present immunofluorescence (IF) images against TrkB and pTrkB in the first figure. While they mention in the materials and methods section that these antibodies were generated for this study, there is no proof of their specificity. It should be noted that most commercial antibodies labeled as anti-TrkB recognize the extracellular domain of all TrkB isoforms. There are indications in the literature that pathological and excitotoxic conditions change the expression levels of TrkB-Fl and TrkB-T1. Therefore, it is necessary to demonstrate which isoform of TrkB the authors are showing as increased under their conditions. Similarly, it is essential to prove that the new anti-p-TrkB antibody is specific to this Trk receptor and, unlike other commercial antibodies, does not act as an anti-phospho-pan-Trk antibody.

Thank you for your kind concern.

Human TrkB has 7 isoforms and predicted Mw ranges from 35 to 93kDa. It has 11 potential N-glycosylation sites. The given antibody (isotype: Mouse IgG2a, κ) has been shown to interact with SHC1, PLCG1 and/or PLCG2, SH2B1 and SH2B2, NGFR, SH2D1A, SQSTM1 and KIDINS220, FRS2.

And, sorry for the misunderstanding and text mistake. We procured all the antibodies from the market using proven products, and didn’t check any specific isoform. We have mentioned the details of antibodies and reagents in the chemicals section of the methodology.

(4) I believe this initial conclusion could be significantly strengthened, without opening up other interpretations of the results, by demonstrating the specificity of the antibodies via Western blot (WB), both in the presence and absence of BDNF and other neurotrophins, NGF, and NT-3. Additionally, using WB could help reinforce the quantification of fluorescence intensity presented by the authors in Figure 1. It's worth noting that the authors fixed the cells with 4% PFA for 2 hours, which can significantly increase cellular autofluorescence due to the extended fixation time, favoring PFA autofluorescence. They have not performed negative controls without primary antibodies to determine the level of autofluorescence and nonspecific background. Nor have they indicated optimizing the concentration of primary antibodies to find the optimal point where the signal is strong without a significant increase in background. The authors also do not mention using reference markers to normalize specific fluorescence or indicating that they normalized fluorescence intensity against a standard control, which can indeed be done using specific signal quantification techniques in immunocytochemistry with a slide graded in black-and-white intensity controls. From my experience, I recommend caution with interpretations from fluorescence quantification assays without considering the aforementioned controls.

Thank you for your insightful comments. We have now included a negative control image in the revised Figures. This control confirms that the observed fluorescence signal is specific and not due to autofluorescence or nonspecific background. In our lab, we have been using these antibodies and already optimized the concentration to use in certain cell types. Additionally, we followed the manufacturer’s recommended antibody concentration and protocol throughout our experiments to ensure an optimal signal-to-noise ratio.

We agree that extended fixation with 4% PFA may increase autofluorescence; however, including negative controls helps account for this effect. We also ensured consistent imaging parameters and applied the same exposure settings across all samples to allow for a valid comparison of fluorescence intensity. We appreciate your emphasis on careful quantification and have clarified these methodological details in the revised Methods section.

(5) In Figure 2, the authors determine the expression levels of TrkA and TrkB using qPCR. Although they specify the primers used for GAPDH as a control in materials and methods, they do not indicate which primers they used to detect TrkA and TrkB transcripts, which is essential for determining which isoform of these receptors they are detecting under different stimulations. Similarly, I recommend following the MIQE guidelines (Minimum Information for Publication of Quantitative Real-Time PCR experiments), so they should indicate the amplification efficiency of their primers, the use of negative and positive controls to validate both the primer concentration used, and the reaction, the use of several stable reference genes, not just one.

We appreciate the reviewer’s suggestion regarding the specificity of primers and the amplification efficiency. In response, we have now included the primer sequences used for detecting TrkA and TrkB transcripts in the revised Materials and Methods section (Quantitative real-time PCR analysis of odontogenic differentiation marker gene expression in dental pulp stem cells). This ensures clarity on which isoforms of these receptors were assessed under different conditions. We also acknowledge the importance of following MIQE guidelines, and we got the primer provided by Integrated DNA Technologies with standard desalting purification and guaranteed yield.

(6) Moreover, the authors claim they are using the same amounts of cDNA for qPCRs since they have quantified the amounts using a Nanodrop. Given that dNTPs are used during cDNA synthesis, and high levels remain after cDNA synthesis from mRNA, it is not possible to accurately measure cDNA levels without first cleaning it from the residual dNTPs. Therefore, I recommend that the authors clarify this point to determine how they actually performed the qPCRs. I also recommend using two other reference genes like 18S and TATA Binding Protein alongside GAPDH, calculating the geometric mean of the three to correctly apply the 2^-ΔΔCt formula.

Thank you for your kind concern. We agree that residual dNTPs from cDNA synthesis could impact the accuracy of cDNA quantification. To address this, we have used the commercially available and guaranteed kit. The kit used is mentioned in Materials and Methods. We will definitely consider using 18S and TATA Binding Protein alongside GAPDH in our future studies. For now, we request you consider the results generated against GAPDH control.

(7) Similarly, given that the newly generated antibodies have not been validated, I recommend introducing appropriate controls for the validation of in-cell Western assays.

We apologize for the text mistake. Antibodies were procured commercially and not generated. We have corrected the sentence.

(8) The authors' conclusion that TrkB levels are minimal (Figure 2E) raises questions about what they are actually detecting in the previous experiments might not be the TrkB-Fl form. Therefore, it is essential to demonstrate beyond any doubt that both the antibodies used to detect TrkB and the primers used for qPCR are correct, and in the latter case, specify at which cycle (Ct) the basal detection of TrkB transcripts occurs. Treatment with TNF-alpha for 14 days could lead to increased cell proliferation or differentiation, potentially increasing overall TrkB transcript levels due to the number of cells in culture, not necessarily an increase in TrkB transcripts per cell.

Thank you for your comments. We appreciate your kind concerns. Here, we are trying to demonstrate that TrkB gets activated in inflammatory conditions. We have also provided the details on primers and antibodies. We have used commercial antibodies and qPCR primers, and they have been extensively validated with previous publications. The efficiency and validation of qPCR primers were provided by a company.

Moreover, we used the minimal concentration of TNF-alpha twice a week, and before using it, we did preliminary experiments to determine whether it affected any experimental condition.

(9) Overall, there are reasonable doubts about whether the authors are actually detecting TrkB in the first three images, as well as the phosphorylation levels and localization of this receptor in the cells. For example, in Figure 3 A to J, it is not clear where TrkB is expressed, necessitating better resolution images and a magnified image to show in which cellular structure TrkB is expressed.

Thank you for your comment. Here, we aimed to show the expression of TrkB receptors in inflamed/infected pulp, especially in minority-distributed DPSCs. TrkB is present on the cell membrane and perinuclear region. We have provided a single-cell (magnified) image in the figure for better clarification.

(10) In Figure 4, the authors indicate they have generated cells overexpressing BDNF after recombination using CRISPR technology. However, the WB they show in Figure 4B, performed under denaturing conditions, displays a band at approximately 28kDa. This WB is absolutely incorrect with all published data on BDNF detection via this technique. I believe the authors should demonstrate BDNF presence by showing a WB with appropriate controls and BDNF appearing at 14kDa to assume they are indeed detecting BDNF and that the cells are producing and secreting it. What antibodies have been used by the authors to detect BDNF? Have the authors validated it? There are some studies reporting the lack of specificity of certain commercial BDNF antibodies, therefore it is necessary to show that the authors are convincingly detecting BDNF.

Dear reviewer, thank you for your kind concern. Firstly, we apologize for the inconvenience.

Pro-BDNF is produced as a 32-kDa precursor that undergoes N-glycosylation and glycosulfation on residues located within the pro-domain of the precursor. N-terminal cleavage of the precursor generates mature BDNF and a minor truncated form of the precursor (28 kDa) that arises by a different processing mechanism than mature BDNF. The precursor undergoes N-terminal cleavage within the trans-Golgi network and/or immature secretory vesicles to generate mature BDNF (14 kDa).

We checked our data and band size, and it shows a mistake in recognizing ladder size. It is actually a 14kDa band which has been shown. The labeling has been revised in the figure, and the actual blot with a ladder has been shown below for clarification. Similarly, our data focused on the fact that the observed cellular effects are more consistent with BDNF/TrkB-mediated pathways, which are known to promote survival and differentiation.

(11) While the RNA sequencing data indicate changes in gene expression in cells treated with TNFalpha+CTX-B compared to control, the authors do not show a direct relationship between these genetic modifications with the rest of their manuscript's argument. I believe the results from these RNA sequencing assays should be put into the context of BDNF and TrkB, indicating which genes in this signaling pathway are or are not regulated, and their importance in this context.

Thank you for your concern. In order to identify the underlying mechanism, we ought to locate certain transcriptional factors interacting with the TrkB/BDNF signaling, leading to differentiation and dentinogenesis. Therefore, we treated it with a TrkB inhibitor.

Earlier, we showed only the combined results and mentioned the interaction between TNFα and TrkB. We have included the results from TNFα alone and combined them with CTX-B for better comparison (Please refer to Figure 8). Figure 8C1 clearly shows the reversal of certain factors with the treatment of TrkB inhibitor compared to figure 8C with TNFα alone treated group. We agree that the precise role of CTX-B in modulating TrkB signaling requires further clarification. We have now included this point in the revised discussion while working on this aspect. In a parallel study, we are trying to dig deep, especially the TCF family, as they have been documented to interact indirectly with BDNF and TrkB.

**Recommendations for the authors:**

**Reviewer #1 (Recommendations for the authors):**
Some minor textual issuesLine 120: It is obvious that TNFα stimulation caused significant phosphorylation of TrkB (p < 0.01) compared to TrkA (p < 0.05).

Thank you for noticing the typo. The sentence has been corrected.

The authors should consider rewording this sentence - I do not understand the intended meaning.Line 126: pronounced peak at 10 ng/mL. I am not convinced there is a peak. Looks like a plateau to me. To call it a peak one would have to show that the values at 10 ng/ml and 20 ng/ml are statistically different.

We meant here the peak compared to 0.1 and 1ng/mL concentration and not compared to 20 ng/mL. The sentence has been elaborated accordingly.

**Reviewer #3 (Recommendations for the authors):**
The authors should show how they have validated the specificity of all the used antibodies as well as the efficiency and specificity of their qPCR data.

We procured the commercially available antibodies (all of them have been extensively validated with previous publications) and also performed negative controls (provided in revised figures). We frequently used Western blot and validate it with band size. Primer sequences are also provided in the revised manuscript. We checked its specificity with R^2^ of Standard Curve ≥ 0.98 and the single peak of melting curves. We edited accordingly in line 263.

Once again, we thank all of you for your efforts in evaluating our study. It really helped us improve the quality of the manuscript. We hope all the queries have been answered and the revised manuscript is acceptable.